# Should Under-parameterized Student Networks Copy or Average Teacher Weights?

**Berfin Şimşek**
NYU*
bs3736@nyu.edu

**Amire Bendjeddou**
EPFL
amire.bendjeddou@epfl.ch

**Wulfram Gerstner**
EPFL
wulfram.gerstner@epfl.ch

**Johanni Brea**
EPFL
johanni.brea@epfl.ch

## Abstract

Any continuous function $f^*$ can be approximated arbitrarily well by a neural network with sufficiently many neurons $k$. We consider the case when $f^*$ itself is a neural network with one hidden layer and $k$ neurons. Approximating $f^*$ with a neural network with $n < k$ neurons can thus be seen as fitting an under-parameterized "student" network with $n$ neurons to a "teacher" network with $k$ neurons. As the student has fewer neurons than the teacher, it is unclear, whether each of the $n$ student neurons should copy one of the teacher neurons or rather average a group of teacher neurons. For shallow neural networks with erf activation function and for the standard Gaussian input distribution, we prove that "copy-average" configurations are critical points if the teacher's incoming vectors are orthonormal and its outgoing weights are unitary. Moreover, the optimum among such configurations is reached when $n - 1$ student neurons each copy one teacher neuron and the $n$-th student neuron averages the remaining $k - n + 1$ teacher neurons. For the student network with $n = 1$ neuron, we provide additionally a closed-form solution of the non-trivial critical point(s) for commonly used activation functions through solving an equivalent constrained optimization problem. Empirically, we find for the erf activation function that gradient flow converges either to the optimal copy-average critical point or to another point where each student neuron approximately copies a different teacher neuron. Finally, we find similar results for the ReLU activation function, suggesting that the optimal solution of underparameterized networks has a universal structure.

## 1 Introduction

A shallow neural network with a single hidden layer of a large number $k$ of neurons can approximate any continuous function $f^*$ arbitrarily well on a compact subset of the input space [1]. We consider a related problem, where the function $f^*$ itself is a neural network with a large number $k$ of neurons, and its approximation is a smaller network with $n < k$ neurons. In other words, we fit an under-parameterized "student" network with $n$ neurons to a "teacher" network with $k$ neurons. As the student has fewer neurons than the teacher, it cannot perfectly match the teacher. In the configuration with the lowest loss, where the approximation error is smallest, one may expect that the incoming and outgoing weights of a student neuron are either identical to those of a teacher neuron or that they are aligned with the weights of a group of teacher neurons, but it is unclear what the optimal configuration is.

---

*Previous address: EPFL.

37th Conference on Neural Information Processing Systems (NeurIPS 2023).

To answer the question of whether student neurons should "copy" or "average" teacher neurons, and more generally to shed light on the loss landscape of under-parameterized neural networks, we study the theoretically tractable setup with standard Gaussian input data and teacher networks with orthogonal incoming vectors. First, we re-parameterize the loss in terms of interactions between pairs of neurons, similar to [2, 3], and we re-formulate the original optimization problem as a constrained optimization problem. The interactions between neurons can be written as a function expressed in terms of the standard deviation and correlation of two Gaussian random variables, with explicit formulas for the erf and ReLU activation functions [2–4]. Next, we prove several properties of the most extremely under-parameterized student network with a single neuron $n = 1$, extending thus the important work of [5–7]. For many commonly used activation functions, we prove for the network with a single hidden neuron that the optimal solution is the only non-trivial critical point of the loss function up to symmetries and is achieved when the incoming vector of the one-neuron student reaches a configuration that can be interpreted as a damped average of all incoming teacher weights.

The proof relies on identifying the critical points of the constrained optimization problem and showing that the common activation functions satisfy the assumptions. We rely in particular on the derivative rule of the interaction function which comes as a pleasant consequence of Stein's Lemma [8] instead of the Hermite basis expansion which is a commonly used technique [9–14]. For the erf and ReLU activation functions we derive additionally a closed-form solution of the optimization problem for $n = 1$. Next, we investigate "copy-average" configurations of students with $n > 1$ neurons, where some student neurons copy teacher neurons and other student neurons average sub-groups of teacher neurons, in the sense that they are at the optimal one-neuron solution for the given sub-group of teacher neurons. Our particular contributions are:

- We propose a constrained optimization formulation of the standard minimization problem in the weight-space in terms of the *interaction function* (Section 3). The interaction function is a natural generalization of the dual activation [15].

- Applying the constrained optimization formulation for $n = 1$, we prove that the incoming vector of the student lies in the span of the incoming vectors of an orthogonal teacher network (Proposition 4.1). For a broad class of activation functions, we prove that the incoming vector aligns with the average of the teacher's incoming vectors for the "unit-orthonormal" teacher network (Theorem 5.1). Using the derivative rule of the interaction function (Lemma F.1), we show that common activation functions such as erf, softplus, tanh, and ReLU satisfy this property (Lemma F.2 and Corollary G.5).

- Assuming a unit-orthonormal teacher network and erf activation function, we prove that the concatenation of critical points of single neurons (of the student network) each approximating a teacher subnetwork is a *copy-average* critical point (Theorem 4.2).

- Assuming a unit-orthonormal teacher network and erf activation function, we prove that the optimal copy-average (CA) configuration is such that $n - 1$ student neurons each copy a teacher neuron and the $n$-th student neuron approximates optimally the sum of the remaining teacher neurons (Theorem 5.5; see also Fig. 1, top row). Empirically, we find that the gradient flow converges to an optimal-CA point for all seeds when $n < \gamma_1 k$ with a fixed $\gamma_1$ near $0.46$ (Figure 4).

- Surprisingly, we find empirically three regimes of training via gradient flow (GF)[2] for under-parameterized networks (Figure 4): (i) for $n < \gamma_1 k$, GF converges to an optimal-CA point for all seeds, (ii) for $n > \gamma_2 k$ with a fixed $\gamma_2$ near $0.6$, GF converges to a point that we call *perturbed-n-copy* for all seeds, (iii) for $\gamma_1 k < n < \gamma_2 k$, GF converges to either an optimal-CA point or a perturbed-n-copy point. Therefore, as the under-parameterized network grows larger, the solution found with gradient flow where the weights are initialized randomly with a fixed standard deviation changes qualitatively. The code to reproduce these findings is available on GitHub, and we refer to Appendix C for details.

## 1.1 Related Work

The teacher-student setup has been extensively used in the literature to study the evolution of gradient flow trajectories and of the generalization error [2, 3, 17–20]. This series of work gives insight into

---

[2]We use a numerical ODE solver for multi-layer networks [16] to simulate the gradient flow in this paper. All "solutions", which are the points at which gradient flow converges, have a gradient norm of at most $5 \cdot 10^{-8}$.

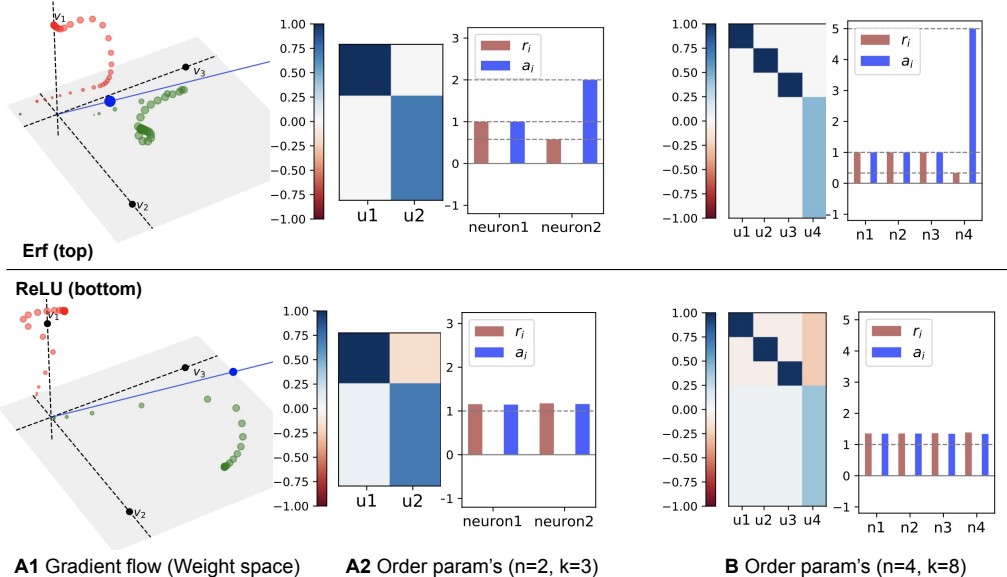

**A1** Gradient flow (Weight space)    **A2** Order param's (n=2, k=3)    **B** Order param's (n=4, k=8)

Figure 1: *The gradient flow converges to the copy-average optimum point for erf activation (top), or nearby for ReLU activation (bottom): the first $n-1$ neurons copy one teacher neuron each; the $n$-th neuron takes an average of the remaining teacher neurons.* The teacher network is unit-orthonormal, i.e. $f^*(x) = \sum_{j=1}^{k} \sigma(v_j \cdot x)$ where $v_j \in \mathbb{R}^d$'s are orthonormal, and $d = k+1$. **A1** The gradient flow trajectory is shown in the weight space for $n=2, k=3$: the positions of the circles (red and green) represent incoming vector $w_i$ projected down to the span of $v_1, v_2, v_3$ and the sizes of the circles represent outgoing weights $a_i$. The blue circle represents the one-neuron solution (the position shows $w^*$, the size shows $a^*$). **A2** Same setting, the weight-space parameters at convergence are mapped to the order-parameter space; $u_i = (u_{i1}, ..., u_{ik})$ where $u_{ij}$ represents the normalized dot product between $w_i$ and $v_j$ and $r_i = \|w_i\|$. **B** Order parameters shown at convergence for $n=4, k=8$. For erf (top) the point at convergence is exactly an $(n-1)$-copy-1-average point, whereas for ReLU, it is perturbed away from this configuration. Neurons are reordered for clarity.

the solution found at convergence, however, they rely on numerically integrating the equations of dynamics. Tian [5] gives convergence guarantees for ReLU activation function, however, their method only works for *one* student and *one* teacher neuron. Xu and Du [21] recently gave the convergence rates for *multiple* student neurons for the case of *one* teacher neuron as a prototypical setup for overparameterization. These convergence guarantees were extended to broad input distributions [7, 22] and finite training data [6, 23]. We give the analytical formula of the optimal solution and its generalization error for *one* student neuron and unit-orthonormal teacher network with *multiple* neurons for erf and ReLU activation functions and a partial characterization for a broader class of activation functions without relying on the analytic formula of the loss.

The studies cited above showed positive results for a single-neuron teacher or a unit-orthonormal teacher. However, even for settings where the teacher has only a few neurons, hard teachers can be constructed in the sense that the student fails to find a zero-loss solution for a certain fraction of random initializations [24–26]. Moreover, for medium-scale problems, gradient flow often converges to 'non-zero loss' solutions [27–29]. Arjevani and Field [30] characterized some families of local minima using symmetries, for the ReLU activation function and unit-orthonormal teacher network. In this paper, we similarly characterize, for the case that the student has a smaller size than the teacher, a large family of 'copy-average' critical points, but for the erf activation function. Our approach focuses on the important regime of under-parameterized networks which is relevant for the superposition of features [31] and for the distillation of large networks into smaller ones [32, 33].

There is a large history of approximation theory of neural networks that give universal guarantees on the approximation error, e.g. [1, 34–36]. However, these works focus on rates of convergence and provide neither a formula nor an approximation for the error. In this paper, we make a conjecture for the *exact formula* of the approximation error of under-parameterized student networks which we support both theoretically and numerically.

## 2 Setup

**Neural network:** Consider a two-layer (student) network function $f : \mathbb{R}^d \to \mathbb{R}$ with $n$ neurons

$$f(x) = \sum_{i=1}^{n} a_i \sigma (w_i \cdot x) \tag{1}$$

where $w_i \in \mathbb{R}^d$ is the incoming vector, $a_i \in \mathbb{R}$ is the outgoing weight of neuron $i$, and the activation function $\sigma$ is twice differentiable unless it is specified to be ReLU, i.e. $\sigma_{\text{relu}}(x) = \max(0, x)$, and the dot marks the scalar product. $P = n(d + 1)$ is the number of parameters.

**Parameter vector:** The parameter vector is represented as

$$\theta = (w_1, a_1) \oplus \ldots \oplus (w_n, a_n) \in \mathbb{R}^P \tag{2}$$

where $\oplus$ denotes the concatenation of two vectors into one vector. We use the notation $\oplus$, since the network function can be seen as a sum of its hidden neurons. Sometimes $\theta$ is written explicitly in the network function $f(x|\theta) = f(x)$.

**Loss function:** We assume that the input distribution is a standard $d$-dimensional Gaussian $\mathcal{D} = \mathcal{N}(0, I_d)$. The target function is denoted by $f^* : \mathbb{R}^d \to \mathbb{R}$. Using the square cost, the loss function $L : \mathbb{R}^P \to \mathbb{R}$ (also known as the risk or the generalization error) is defined as

$$L(\theta) = \mathbb{E}_{x \sim \mathcal{D}} \left[ (f(x|\theta) - f^*(x))^2 \right]. \tag{3}$$

**Orthogonal teacher network:** We assume that the target function is a neural network (also known as the teacher network or a multi-index model)

$$f^*(x) = \sum_{j=1}^{k} b_j \sigma(v_j \cdot x) \tag{4}$$

where its outgoing weights are non-zero and its incoming vectors $v_1, \ldots, v_k \in \mathbb{R}^d$ are orthogonal to each other, that is, $v_i \cdot v_j = 0$ for $i \neq j$. This implies that the input dimension satisfies $d \geq k$. Following [27, 30, 37], we particularly focus on the *unit-orthonormal* teacher network where the outgoing weights are all one, that is $b_j = 1$, and the incoming vectors have unit norm, i.e. $v_i \cdot v_j = \delta_{ij}$.

**Optimal loss:** We study the optimal solution(s) of the following non-convex optimization problem

$$L^{n,k}(\oplus_{i=1}^{n}(a_i, w_i)) = \mathbb{E}_{x \sim \mathcal{D}} \left[ \left( \sum_{i=1}^{n} a_i \sigma(w_i \cdot x) - \sum_{j=1}^{k} b_j \sigma(v_j \cdot x) \right)^2 \right]. \tag{5}$$

for under-parameterized (student) networks, i.e. $n < k$, and orthogonal teachers. For $n \geq k$ neurons, the network can copy all teacher neurons and set the outgoing weights of the remaining neurons to zero, therefore the optimal loss is trivially zero. If the teacher is unit-orthonormal, then all of its neurons contribute equally; hence the optimal loss is determined by $n$ and $k$ only and denoted by $L^*(n, k)$. If the student neural network has one neuron we use the notation $L^*(k) := L^*(1, k)$.

## 3 Foundations & Constrained Optimization Formulation

In this section, we introduce a constrained optimization problem that is a reformulation of the minimization problem in Eq. 5. This formulation allows us to show that the incoming vector of any non-trivial critical point of the one-neuron network is in the span of the teacher's $k$ orthogonal (or potentially even non-orthogonal, see Appendix Remark D.1) incoming vectors (see Proposition 4.1). We give the exact solution in the case of a unit-orthonormal teacher (see Corollary 5.2 and Corollary G.5).

Using the linearity of expectation, the loss function in Eq. 5 can be expanded as a weighted sum of the following Gaussian integral terms

$$\mathbb{E}_{x \sim \mathcal{D}}[\sigma(V_1 \cdot x)\sigma(V_2 \cdot x)]$$

where $V_1$ and $V_2$ represent two arbitrary vectors of student and teacher networks such as $(w_i, w_j)$ or $(w_i, v_j)$. As both $V_1 \cdot x$ and $V_2 \cdot x$ are centered Gaussian random variables, the above expectation can be expressed in terms of the covariance of the two-dimensional Gaussian

$$\mathbb{E}_{x \sim \mathcal{D}} \begin{bmatrix} (V_1 \cdot x)^2 & (V_1 \cdot x)(V_2 \cdot x) \\ (V_1 \cdot x)(V_2 \cdot x) & (V_2 \cdot x)^2 \end{bmatrix} = \begin{bmatrix} r_1^2 & r_1 r_2 u \\ r_1 r_2 u & r_2^2 \end{bmatrix}$$

where $r_i := \|V_i\|$ for $i = 1, 2$ is the $\ell_2$-norm and, assuming $r_i > 0$, $u := V_1 \cdot V_2 / (r_1 r_2)$ is the correlation. The covariance entries $Q_{ii} = r_i^2, Q_{12} = r_1 r_2 u$ have been used to study the gradient flow trajectories [2, 3, 24]. We prefer the parametrization with $r_i$ and $u$ as it enables us to make the positive definiteness constraint explicit, i.e.

$$|u| = \frac{|V_1 \cdot V_2|}{r_1 r_2} \leq 1, \tag{6}$$

due to the Cauchy-Schwarz inequality. We introduce the *interaction function* $g_\sigma : \mathbb{R}_{\geq 0}^2 \times [-1, 1] \to \mathbb{R}$

$$g_\sigma(r_1, r_2, u) = \mathbb{E}_{(x_1, x_2) \sim \mathcal{N}(0, \Sigma)}[\sigma(r_1 x_1) \sigma(r_2 x_2)] \quad \text{with } \Sigma = \begin{bmatrix} 1 & u \\ u & 1 \end{bmatrix}, \quad r_1, r_2 > 0, \tag{7}$$

to express the Gaussian integral terms. Note that $u$ is not well-defined if one of the norms is zero. Extending the formula above, for the case w.l.o.g. $r_2 = 0$, we define

$$g_\sigma(r_1, 0, u) := \mathbb{E}_{x \sim \mathcal{N}(0,1)}[\sigma(r_1 x)]\sigma(0)$$

for all $u \in [-1, 1]$. In this paper, we consider the activation functions satisfying the following.

**Assumption 3.1.** *For all $r_1, r_2 > 0$ and $u \in (-1, 1)$, we assume that the interaction function $g_\sigma$ satisfies either the first or both of the following properties*

$$(i) \ \frac{d}{du} g_\sigma(r_1, r_2, u) > 0, \qquad (ii) \ \frac{d^2}{du^2} g_\sigma(r_1, 1, u)u < \frac{d}{du} g_\sigma(r_1, 1, u). \tag{8}$$

To check whether a specific activation function satisfies the above properties, we mainly rely on Lemma F.1 which gives us the rule for the partial derivative of $g_\sigma$ with respect to the correlation

$$\frac{d}{du} g_\sigma(r_1, r_2, u) = r_1 r_2 \mathbb{E}[\sigma'(r_1 x)\sigma'(r_2 y)]. \tag{9}$$

Hence, if $\sigma$ is monotonic (increasing or decreasing)[3], the integrand on the right-hand side is positive; satisfying Assumption 3.1 (i). The ReLU activation function $\sigma_{\text{relu}}(x) = \max(0, x)$ also satisfies it because of the known analytical expression of the interaction [4, 27]

$$g_{\text{relu}}(r_1, r_2, u) = r_1 r_2 h(u) \quad \text{where } h(u) = \frac{1}{2\pi}\left(\sqrt{1 - u^2} + (\pi - \arccos(u))u\right).$$

Checking Assumption 3.1 (ii) for a given activation function is delicate. We rely on it in Section 5.

Using the interaction function, the loss function can be expressed in terms of the *order parameters:*

- norms of the incoming vectors of the student $r_i = \|w_i\|$,
- correlations between the incoming vectors of the student and teacher $u_{ij} = w_i \cdot v_j / (r_i \|v_j\|)$,
- correlations between the incoming vectors of the student $\rho_{ii'} = w_i \cdot w_{i'} / (r_i r_{i'})$;

where we assumed $r_i > 0$ for all $i \in [n]$. The constrained optimization formulation is possible for general non-orthogonal teacher networks (see Remark D.1 in the Appendix). For the sake of simplicity, we formulate here the constrained optimization problem for the case of orthogonal teachers and reformulate the objective in Eq. 5 as

$$\text{minimize} \ \sum_{i=1}^n a_i^2 g_\sigma(r_i, r_i, 1) + 2 \sum_{i \neq i'} a_i a_{i'} g_\sigma(r_i, r_{i'}, \rho_{ii'}) - 2 \sum_{i=1}^n \sum_{j=1}^k a_i b_j g_\sigma(r_i, \|v_j\|, u_{ij}) + C$$

$$\text{subject to } \|u_i\| \leq 1, \ r_i \geq 0, \quad \text{for all } i \in [n],$$

$$\left| \rho_{ii'} - u_i \cdot u_{i'} \right| \leq \sqrt{1 - \|u_i\|^2} \sqrt{1 - \|u_{i'}\|^2}, \quad \text{for all } i \neq i' \in [n]; \tag{10}$$

---

[3] Increasing (or decreasing) mean strictly increasing (or decreasing) everywhere in this paper.

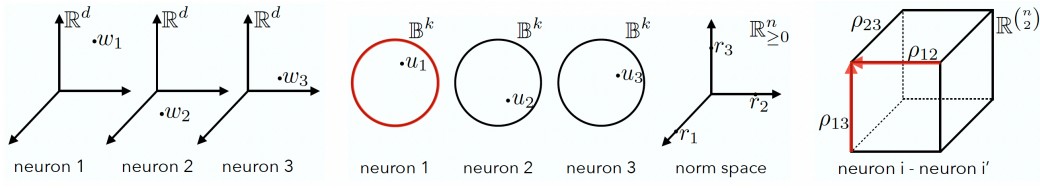

**A** Weight-space      **B1** Primary domain of the order parameters      **B2** Secondary domain

Figure 2: *Cartoon representation of the mapping of a student with three neurons from the weight space **A** $\mathbb{R}^{nd}$ to order parameter space **B1-B2**. The mapping between the outgoing weights is an identity mapping hence not shown.* **A** *Each axis shows the direction of weights $v_i$ of one teacher neuron ($k \geq 3$).* **B1** *Each incoming vector $w_i \in \mathbb{R}^d$ is first transformed into $(r_i, w_i/r_i)$ and then $w_i/r_i$ is projected onto the span of the teacher's incoming vectors, yielding the student-teacher correlation vector $u_i = (u_{i1}, ..., u_{ik})$.* **B2** *The student-student correlations $\rho_{ii'}$ are in general free parameters bounded in between $u_i \cdot u_{i'} \pm \sqrt{1 - \|u_i\|^2}\sqrt{1 - \|u_{i'}\|^2}$ hence the box constraint. An activated constraint, w.l.o.g. $u_1 \in \mathbb{S}^{k-1}$, gives a vanishing $\pm$ term for the interval of correlation $\rho_{1i}$ for all $i \neq 1$, hence they are no longer free (shown in red). In the case $d = k$, all $u_i$ are on the hypersphere due to the problem geometry, hence the correlations $\rho_{ii'}$ are fixed and not free (see Appendix D.1).*

where $u_i = (u_{i1}, ..., u_{ik})$ and $C = \mathbb{E}_{x \sim \mathcal{D}}[f^*(x)^2]$. The constraints in Eq. 10 give tighter bounds than simply bounding correlations with the help of Eq. 6. See Appendix D for the derivation of the constraints and Fig. 2 for a schematic.

The objective above is *exact* for $n = 1, 2$, in the sense that its optimal solution is equivalent to the optimal solution in the weight space, since the mapping from the weight-space to the order space is invertible. However, it is a *relaxation* for $n \geq 3$, since there are order-parameter configurations in the domain (see Figure 2) that do not correspond to any weight-space configuration (see Appendix D.3 for a construction). It seems possible to overcome this gap by considering the geometry of the angles between $n \geq 3$ incoming vectors to tighten the constraints between student-student correlations.

## 4 Copy-Average Critical Points

In this section, we identify a new family of critical points by 'combining' critical points of one-neuron networks for the unit-orthonormal teacher and the erf activation function. We first show that in a network with $n = 1$ student neuron, for any "non-trivial" critical point, that is $w^* \neq 0$ and $a^* \neq 0$, the incoming vector $w^*$ is in the span of the teacher's incoming vectors (Proposition 4.1). Applying this proposition to the special case of the erf activation function, we show that the concatenation of such critical points is also a critical point for multi-neuron networks (Theorem 4.2).

**Proposition 4.1.** *Assume that $f^*$ is an orthogonal teacher network (Eq. 4) of width $k$. If the activation function satisfies Assumption 3.1 (i), any non-trivial critical point $\theta^* = (w^*, a^*)$, i.e. $\nabla L^{1,k}(\theta^*) = 0$, $\|w^*\| \neq 0$, $a^* \neq 0$, satisfies that $w^*$ is in the span of the teacher's incoming vectors.*

The proof uses the constrained optimization formulation for $n = 1$ (see Appendix G.1). In short, a critical point mapped to the order parameter space satisfies either $\|u_1\| = 1$ or $\partial_u g_\sigma(r, \|v_j\|, u_{1j}) = 0$ for all $j \in [k]$. Under the Assumption 3.1 (i) these partial derivatives are non-zero, hence $\|u_1\| = 1$. In a recent work [38], the incoming vectors of the student network are also shown to converge to the span of the vectors of the multi-index model using weight-decay.

Finding the optimal solution for the multi-neuron network is challenging. Natural candidates are concatenation of student neurons where each one of them is a critical point of the loss function $L^{1,\ell_i}$ where $\ell_i$ is the number of the subgroup of teacher neurons. More precisely, let us pick a partition $\ell_1 + ... + \ell_n \leq k$ such that $\ell_i \geq 1$, and define $s_m = \sum_{i=1}^{m} \ell_i$ for $m \leq n$ and $s_0 = 0$. We denote a one-neuron critical point by $\theta_i^* = (w_i^*, a_i^*)$, when learning from a part of the teacher network

$$f_i^*(x) = \sum_{j=s_{i-1}+1}^{s_i} \sigma(v_j \cdot x). \tag{11}$$

Since $f_i^*$ is a unit-orthonormal teacher, $w_i^*$ is in the span of $v_{s_{i-1}+1}, ..., v_{s_i}$ due to Proposition 4.1.

We use the term *copy-average* (CA) point or configuration to refer to the concatenation of such one-neuron critical points in the student network with $n$ neurons: if $\ell_i = 1$, the student neuron copies one of the teacher neurons $(v_j, 1)$; if $\ell_i > 1$, it averages a group of teacher neurons in the sense of approximating their sum with one neuron. For odd activation functions, the one-neuron network problems decouple from each other, as the cross-terms $\mathbb{E}[\sigma(w_1 \cdot x)\sigma(w_2 \cdot x)]$ vanish for $w_1 \perp w_2$. For the specific case of erf, we prove that all the copy-average configurations are critical points.

**Theorem 4.2.** *Assume that $\sigma(x) = \sigma_{erf}(x) = \frac{2}{\sqrt{\pi}} \int_0^{\frac{x}{\sqrt{2}}} e^{-t^2} dt$. We pick a copy-average parameter*

$$\theta^* = (w_1^*, a_1^*) \oplus ... \oplus (w_n^*, a_n^*) \tag{12}$$

*where $(w_i^*, a_i^*)$ is a non-trivial critical point when learning from a unit-orthonormal teacher $f_i^*$ with the incoming vectors $v_{s_{i-1}+1}, ..., v_{s_i}$ shown in Eq. 11. Then $\theta^*$ is a critical point of the loss function $L^{n,k}$ where the target function is $f^*(x) = \sum_{j=1}^k \sigma(v_j \cdot x)$.*

In particular, all neurons are equivalent to each other in a unit-orthonormal teacher network. Therefore, the copy-average configurations where $n-1$ student neurons each copy a distinct teacher neuron and the $n$-th student neuron takes an average are also equivalent and called $(n-1)$-copy-1-average, or $(n-1)$-C-1-A in short. Another interesting configuration is where $n$ student neurons each copy a distinct teacher neuron, which is called $n$-copy, or $n$-C in short.

For general activation functions the copy-average parameter vectors are not critical points (see Eq. 56). Nevertheless, we numerically find that the gradient flow converges to similar configurations for the ReLU activation function (see Figure 1, see Appendix C.3 for more experiments).

## 5 Approximation Error of Underparameterized Networks

The target function is assumed to be a unit-orthonormal teacher network in this section. In Subsection 5.1, we show for the one-neuron network that there is a unique non-trivial critical point up to symmetries, which is necessarily the global minimum (Theorem 5.1). Furthermore, we give the analytic expression of the optimal solution and its loss for erf (Corollary 5.2) and ReLU (Corollary G.5) activation functions. In Subsection 5.2, we provide for the under-parameterized student with $n > 1$ neurons the exact loss of copy-average critical points for the erf activation function and show that the $(n-1)$-copy-1-average configurations reach the lowest loss among CA-critical points (see also Appendix E.1 for the combinatorial number of the equivalent copy-average configurations related to the landscape complexity calculations [39]).

### 5.1 One-Neuron Network

Using the constrained optimization formulation in 10, we first prove that at any non-trivial critical point of the one-neuron network, the incoming vector aligns equally with all teacher's incoming vectors for unit-orthonormal teachers for activation functions satisfying Assumption 3.1 (see Theorem 5.1). This is related to the symmetric solution visited during the learning plateaus studied in Saad and Solla [2] for erf activation and in Tian [5] for ReLU activation (see Appendix B for a detailed comparison). Our proof works for a broad class of activation functions and does not use the analytic expression of the interaction function.

**Theorem 5.1.** *Assume that the activation function satisfies Assumptions 3.1 (i) and (ii). At any non-trivial critical point $(w^*, a^*)$ of the loss $L^{1,k}$ for the unit-orthonormal teacher network, the incoming vector satisfies*

$$\frac{w^*}{\|w^*\|} = u^* \sum_{j=1}^k v_j \tag{13}$$

*where $u^*$ is either $\frac{1}{\sqrt{k}}$ or $-\frac{1}{\sqrt{k}}$.*

*Proof Sketch.* There is no student-student interaction term since we have a single neuron; therefore we write $u_j$ instead of $u_{1j}$ and the constrained optimization problem in 10 simplifies to

$$\text{minimize } a^2 g_\sigma(r, r, 1) - 2a \sum_{j=1}^k g_\sigma(r, 1, u_j) + \text{const}, \quad \text{subject to } \|u\| \leq 1, r \geq 0. \tag{14}$$

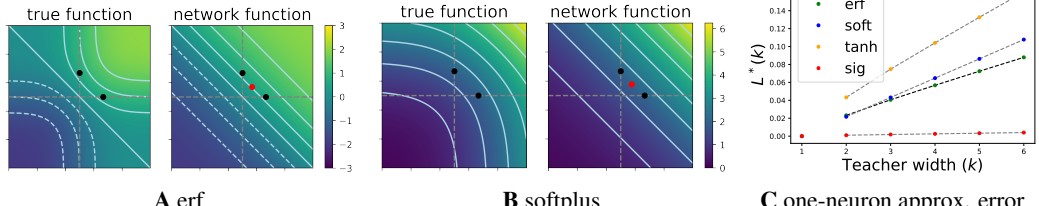

| true function | network function | | true function | network function | |
| **A** erf | | | **B** softplus | | **C** one-neuron approx. error |

Figure 3: *One-neuron network solutions.* **A** Network output (color coded) as a function of input in $d = 2$ for (left) a unit-orthonormal network with $k = 2$ neurons (incoming vectors $v_1$ and $v_2$ are shown as black dots) and (right) the student network function generated by the optimal solution (incoming vector shown in red) for the erf activation function. **B** Same for the softplus activation function. **C** Approximation error of a student with $n = 1$ neurons as a function of the number of $k$ teacher neurons. For large $k$, the approximation error for $n = 1$ grows near-linearly for the differentiable activation functions studied in this paper (erf, sigmoid, tanh, and softplus with $\beta = 1$); however the growth is quadratic for ReLU (see Appendix Corollary G.5).

From Proposition 4.1, we have that $\|u\| = 1$ for any non-trivial critical point. Therefore, the constraint of 14 on the correlations $u = (u_1, ..., u_k)$ is satisfied. The mapping of any non-trivial critical point to the order-parameter space is a critical point of the Lagrangian loss (see Appendix Lemma G.2). Hence every $u_j$ satisfies

$$-2a\frac{d}{du_j}g_\sigma(r, 1, u_j) + 2\lambda u_j = 0 \tag{15}$$

for fixed $(r, a)$. Assumption 3.1-(ii) implies that $\frac{1}{u}\partial_u g_\sigma(r, 1, u)$ is one-to-one hence all $u_j$ are equal. *End of Proof Sketch.*

We show in Lemma F.2 that the interactions of the common activation functions such as erf, tanh, sigmoid, and softplus (respectively)

$$\sigma_{\text{erf}}(x) = \frac{2}{\sqrt{\pi}}\int_0^{\frac{x}{\sqrt{2}}} e^{-t^2} dt, \ \sigma_{\text{tanh}}(x) = \frac{1 - e^{-x}}{1 + e^{-x}}, \ \sigma_{\text{sig}}(x) = \frac{1}{1 + e^{-x}}, \ \sigma_{\text{soft}}^\beta(x) = \frac{1}{\beta}\log(e^{\beta x} + 1),$$

with $\beta \in (0, 2]$ satisfy Assumption 3.1-(ii). The interaction of the ReLU activation function, i.e. $\sigma_{\text{relu}}(x) = \max(0, x)$ also satisfies Assumptions 3.1 (with a slight modification in the domain for (ii); see the proof of Corollary G.5).

Thanks to Theorem 5.1, the loss in Eq. 14 can be reduced to a two-dimensional loss in $a$ and $r$, which can be solved explicitly for ReLU (Corollary G.5) and erf.

**Corollary 5.2.** *Assume that the activation function is $\sigma_{\text{erf}}$. The optimal solution $(w^*, a^*)$ is given by*

$$\|w^*\| = \sqrt{\frac{1}{2k - 1}}, \quad a^* = k, \quad \frac{w^*}{\|w^*\|} = \frac{1}{\sqrt{k}}\sum_{i=1}^k v_i,$$

*or, equivalently, by $(-w^*, -a^*)$. The optimal loss is then given by*

$$L_{\text{erf}}^*(k) = \frac{2}{\pi}\Big(k\arcsin\big(\frac{1}{2}\big) - k^2\arcsin\big(\frac{1}{2k}\big)\Big). \tag{16}$$

The proof of Corollary 5.2 is presented in Section G.3. For general activation functions, the two-dimensional loss does not admit an analytical expression. For this case, from the partial derivatives, we obtain a fixed-point equation in $r$ which we solve numerically for the activation functions listed above (see Appendix Section G.2.2). For softplus, specifically, we prove in addition the following

**Theorem 5.3.** *Assume that the activation function is $\sigma_{\text{soft}}^\beta(x)$ with $\beta \leq 2$. The optimal solution $(w^*, a^*)$ satisfies $\|w^*\| \leq 1/\sqrt{k}$ and $a^* \geq k$.*

We use the FKG inequality to prove Theorem 5.3 (see Appendix G.5). Although the proof requires specific properties of softplus, we show that the above bounds hold also for tanh and sigmoid by numerically solving the fixed point equation (Figure 10, also Figure 5). Note that the incoming vector norm is bounded above by $1/\sqrt{k}$, hence $w^*$ is a *damped* average of the teacher incoming vectors.

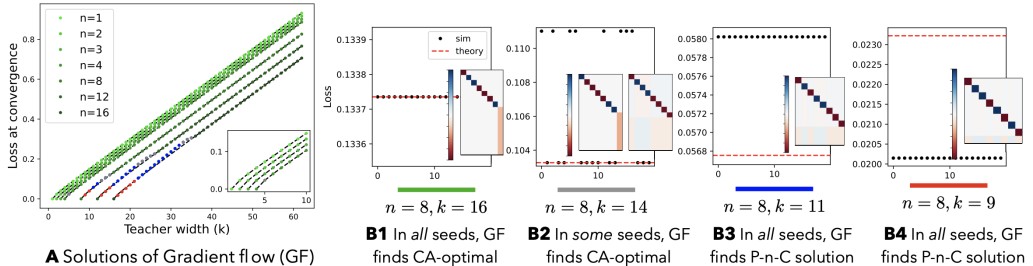

Figure 4: *Under-parameterized student networks of width $n$ with erf activation function learning (via gradient flow) from a unit-orthonormal teacher network of width $k$.* **A** Each dot is the mean error at convergence for 20 seeds of random initializations; black-dashed lines are the theory predictions $L^*_{erf}(k-n+1)$, see Eq. 20. Standard deviations do not show on the figure as they are too small. We identify four regimes indicated by colors (green-gray-blue-red) depending on the type of solution found by gradient flow (GF). In the green regime, GF converges to an optimal $(n-1)$-C-1-A solution for all 20 initializations (Fig. 4-B1). In the gray regime, GF converges either to $(n-1)$-C-1-A solution or to a "Perturbation of the all-copy solution" that we call P-$n$-C (Fig. 4-B2). In the blue and red regimes, for $n > \gamma_2 k$ where $n = 8, 12, 16$ the gradient flow converges to a P-$n$-C solution from all seeds (Fig. 4-B3). Moreover, in the red regime, for $n > \gamma_3 k$ where $n = 8, 12, 16$ and $\gamma_3$ is near 0.75, the P-$n$-C solutions achieve lower loss than the $(n-1)$-C-1-A solutions (Fig. 4-B4). **B1-B4** Examples of loss at convergence (vertical axis) for all 20 different initialization seeds (horizontal axis); theory is shown by the red-dashed horizontal line. Insets show examples of correlation matrices $u_{ij}$ ($k$ lines, $n$ columns) between student and teacher incoming vectors at convergence after reordering neurons. In the gray regime (for ex. B2) the gradient flow converges to either one of the two types of minima with correlations shown in the inset; in the other regimes, it consistently converges to the same minimum up to permutations.

**Remark 5.4.** *We do not impose either of the two reductions that are common in literature: (i) incoming vector $w$ is constrained to be on the unit sphere [9, 11, 14, 40], (ii) the outgoing weight $a$ is constrained to be one [7, 18, 20]. An important step in our analysis is related to the norm $r$ of the incoming vector which we discuss in Appendix Section G.2.2.*

## 5.2 Multi-Neuron Network

In this subsection, we assume that the activation function is erf such that CA-configurations are critical points (see Theorem 4.2). For a student network with $n = 2$ and a partition $(\ell_1, \ell_2)$ we can decompose the loss of a CA critical point as

$$L^*_{erf}(\ell_1) + L^*_{erf}(\ell_2) + L^*_{erf}(0, k - (\ell_1 + \ell_2)) \tag{17}$$

where $L^*_{erf}(0, \ell_0) := \mathbb{E}_{x \sim \mathcal{D}}[f^*_{\ell_0}(x)^2]$ is the error made by a student with vanishing output when representing a reduced unit-orthonormal teacher network with $\ell_0$ neurons. This decomposition is possible because the cross-terms between orthogonal vectors are zero for odd activation functions. Furthermore, for the erf activation function, we show that

$$L^*_{erf}(\ell_1) + L^*_{erf}(0, \ell_0) > L^*_{erf}(\ell_1 + \ell_0) \tag{18}$$

(see the proof of Lemma E.2). Therefore, we should search for the minimum loss configuration among the partitions with $\ell_1 + \ell_2 = k$. Among such partitions, Lemma E.2 shows that the optimum CA-point has the partition $(1, k-1)$. In words, the optimum is a 1-copy-1-average point.

For general $n$, using Lemma E.2 and a small trick, we prove the following.

**Theorem 5.5.** *Consider a unit-orthonormal teacher network $f^*(x) = \sum_{j=1}^k \sigma(v_j \cdot x)$ and the erf activation function. For an under-parameterized student network with $n$ neurons, the minimum-loss copy-average configuration up to permutations (of the student and teacher neurons) is*

$$\theta = (\epsilon_1 v_1, \epsilon_1) \oplus ... \oplus (\epsilon_{n-1} v_{n-1}, \epsilon_{n-1}) \oplus (\epsilon_n w^*_n, \epsilon_n a^*_n) \tag{19}$$

*where $\epsilon_i \in \{\pm 1\}$ and $(w^*_n, a^*_n)$ is given by Corollary 5.2 after substituting $k$ with $k-n+1$.*

See Appendix E.3 for the proof. Because copy-average critical points are not necessarily the only critical points of the loss function for students with $n > 1$, we investigate in simulations, if they are found by gradient flow where the weights are initialized as Gaussian with a fixed standard deviation (see Fig. 4).

Interestingly, gradient flow converges to the CA-optimal solution for all random seeds in a broad regime of under-parameterization (green in Fig. 4). Only when $n > \gamma_1 k$ for $n = 8, 12, 16$ and $\gamma_1 \sim 0.46$, gradient flow converged in some seeds to points close to the $n$-copy critical point. We call these newly found points "perturbed $n$-copy" (P-$n$-C) points. In gray-blue regimes, the P-$n$-C points have higher loss than the optimal CA critical point (Fig. 4).

However, this is not always the case: when the student width is close to the teacher width (low compression regime), the P-$n$-C point has a slightly lower loss the lowest amongst the CA critical points (red in Fig. 4). When $k - n$ is fixed, we found that the $(n - 1)$-C-1-A solution turns from a minimum for small $n$ to a saddle for large $n$ (see App. Fig. 6); which explains why the gradient flow escapes it in this regime and converges to another minimum at a lower loss.

Finally, based on our theory and experiments, we conjecture that there exists a $\gamma_0 \in (0, \gamma_3)$ such that when $n < \gamma_0 k$ and when the activation function is erf, the global optimum of the non-convex loss in Eq. 5 is a $(n-1)$-C-1-A configuration. Therefore, if our conjecture holds, the *exact* approximation error, i.e. the optimal loss, is identical to that of a one-neuron network approximating a teacher with $k-n+1$ neurons and is given by

$$L^*_{\text{erf}}(n, k) = L^*_{\text{erf}}(k-n+1). \tag{20}$$

## 6   Conclusion & Future Directions

We studied the learning of under-parameterized student networks from orthogonal teacher networks for standard Gaussian input data and vanishing thresholds. For erf activation function, we introduced a new family of critical points that arise from the decoupling of the problem into one-neuron networks that can be solved separately. Moreover, the exact parameters of copy-average (CA) critical points are given which can be used to study escape behavior near saddles and to determine convergence of first and second-order optimization algorithms [16, 41].

Furthermore, we showed that the optimal CA point is that $n - 1$ neurons copy teacher neurons and the $n$-th neuron averages the remaining $k - n + 1$ neurons. In simulations, gradient flow converges to a CA-optimal solution for $n < \gamma_1 k$ where $\gamma_1$ is near $0.46$. However, for $n > \gamma_2 k$ where $\gamma_2$ is near $0.6$, we observe another phase where the gradient flow finds a perturbed copy solution. For the ReLU activation function and the onset of under-parameterization, gradient flow converges to qualitatively similar solutions; however, at the crossing point from under-parameterization to over-parameterization (i.e. $n = k$), gradient flow is known to get stuck in spurious local minima [27]. On another note, determining the CA-optimal solution of the two-neuron network plays a critical role in our analysis. Still, there is only little literature on two-neuron networks [5, 42] compared to the well-studied one-neuron case [5–7, 22, 23]. The two-neuron network is possibly the simplest model with interactions between neurons, hence it is important to understand the global minimum and gradient flow dynamics of this challenging problem.

On the practical side, our analysis of under-parameterized networks gives a recipe for how to warm-start smaller neural networks for distilling unit-orthonormal teacher networks. If one desires low compression ($n > \gamma_3 k$), then we recommend initializing the student network in a configuration where each neuron copies a different teacher neuron, to be close to a P-$n$-C point. However, for higher compression, we recommend initializing the student network in a configuration where $n - 1$ neurons are each copied and the $n$-th neuron is initialized as an average neuron to be close to a $(n - 1)$-copy-1-average point. It remains an open question whether this recipe applies to non-idealized scenarios such as non-isotropic input distribution, teacher networks with non-orthogonal incoming vectors, or non-unit outgoing weights. More generally, it is natural to expect that the optimal distillation strategy changes from low compression levels to high compression levels. How exactly and where this change happens is a very intriguing question of theory and practice.

## Acknowledgements

The authors thank Lenka Zdeborová for the discussions and encouragement at the beginning of this project and Clément Hongler for many discussions and valuable feedback. This work was supported by the Swiss National Science Foundation (no. 200020_207426).

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

## A   Summary of Results

Table 1: Summary of Results

| columns=conditions: lines=results: | Orthogonal | UO & erf | UO & $\sigma$ satisfying $g_\sigma$ assumptions |
|---|---|---|---|
| $n = 1$ average is the optimal solution | n.i.g. | yes* | yes |
| $n > 1$ CA points are critical points | n.i.g. | yes | n.i.g. maybe for odd |
| $n > 1$ $(n-1)$-C-1-A is the optimal-CA solution | n.i.g. | yes | n.i.g. maybe for some odd |
| $n = 1$ $w^*$ is in the span of $\{v_1, ..., v_k\}$ | yes | yes* | yes* |

In the table above, UO means unit-orthonormal, n.i.g. stands for 'not in general' and yes* follows as a special case from the results with yes on the same row.

## B   Further Comparison to Literature

In this section, we compare the symmetric solutions found in erf [2] and ReLU networks [5] to our one-neuron solution ($n = 1$). The main difference is that both earlier studies constrain the search space to the symmetric subspace whereas we first prove that the non-trivial critical points are contained in this subspace in Theorem 5.1 for a broad class of activation functions, including erf and ReLU. Solving the low-dimensional loss, we recover the same solution for ReLU and erf as in [2, 5] for unit-orthonormal teachers.

*Symmetric Solution of Saad and Solla [2] for erf activation.* The authors focus on the 'symmetric subspace' parameterized as

$$Q_{ii} = r_i^2 = Q, \quad Q_{ij} = p_{ij} r_i r_j = C, \quad R_{in} = u_{in} r_i = R. \tag{21}$$

In this case, the loss is parameterized by three values, that is $Q, C, R$, hence can be expressed analytically in terms of these values. Solving the fixed point equations, they find the following critical/fixed point (their Eq.22)

$$Q = C = \frac{1}{2k-1}, \quad R = \frac{1}{\sqrt{k(2k-1)}} \tag{22}$$

which implies $r_i = 1/\sqrt{2k-1}$ and $\rho_{ij} = 1$ in our parameterization. This selection of parameters forces all student vectors to be equal therefore reducing the system to a one-neuron network. There are two main improvements in our analysis

1. We prove that student-teacher correlations $u_{ij}$ are equal to each other at a non-trivial critical point, and give necessary conditions on the activation function (Assumption 3.1) to satisfy this property. We show in Lemma F.2 that not only erf but a large class of common activation functions satisfy Assumption 3.1.

2. Our student network has a flexible outgoing weight (shallow neural network) as opposed to a fixed outgoing weight $+1$ (soft-committee machine) in Saad and Solla [2]. It is instructive to compare the generalization errors of the one-neuron network

$$\text{(soft-committee machine)} \ \ L^*_{\text{erf; soft}}(k) = \frac{k}{3} - k^2 \frac{2}{\pi} \arcsin(\frac{1}{2k})$$

$$\text{(shallow network)} \ \ L^*_{\text{erf}}(k) = k \frac{2}{\pi} \arcsin(\frac{1}{2}) - k^2 \frac{2}{\pi} \arcsin(\frac{1}{2k}) \approx k(\frac{1}{3} - \frac{1}{\pi}),$$

which are identical since $\arcsin(0.5) = \pi/6$ (Saad and Solla [2] uses $\epsilon_g(k) = \frac{1}{2} L^*_{\text{erf; soft}}(k)$ that's why there is a factor $0.5$ difference with respect to their Eq. (23)). However, if we set teacher outgoing weights to say $a_t$, the shallow network adapts and reaches the generalization error $a_t^2 L^*_{\text{erf}}(k)$ but the error of the soft-committee machine is

$$L^*_{\text{erf; soft}}(k) = k^2 g(\frac{1}{\sqrt{2k-1}}, \frac{1}{\sqrt{2k-1}}, 1) - 2k^2 a_t g(\frac{1}{\sqrt{2k-1}}, 1, \frac{1}{\sqrt{k-1}}) + a_t^2 k g(1,1,1)$$

$$= O(a_t) + a_t^2 k \frac{1}{3}.$$

which has a worse coefficient $\frac{1}{3}$ compared to $\frac{1}{3} - \frac{1}{\pi}$ as expected.

*Symmetric Solution of Tian [5] for ReLU activation.* The authors focus on a particular two-dimensional subspace $(x, y)$ that allows the specialization of student neurons, namely

$$w_i = x v_i + y \sum_{j \neq i} v_j. \tag{23}$$

In particular, they consider the 'symmetric subspace' $x = y$ which is the case when all student neurons collapse to one neuron, and show that the dynamics converge to the following fixed point

$$x = y = \frac{1}{\pi k}(\sqrt{k-1} - \arccos(\frac{1}{\sqrt{k}}) + \pi). \tag{24}$$

Summing over $k$ neurons then produces the following one-neuron due to the positive homogeneity

$$w^* = \frac{1}{\pi}(\sqrt{k-1} - \arccos(\frac{1}{\sqrt{k}}) + \pi) \sum_{j=1}^{k} v_j.$$

Our formula (Corollary G.5) gives the identical result due to

$$w^* a^* = \frac{k}{h(1)} h(\frac{1}{\sqrt{k}}) \sum_{j=1}^{k} \frac{1}{\sqrt{k}} v_j = \frac{1}{\pi}(\sqrt{k-1} - \arccos(\frac{1}{\sqrt{k}}) + \pi) \sum_{j=1}^{k} v_j.$$

In this case, there is no difference between the optimal solution of the soft-committee machine and the shallow network since ReLU is positive-homogeneous as expected.

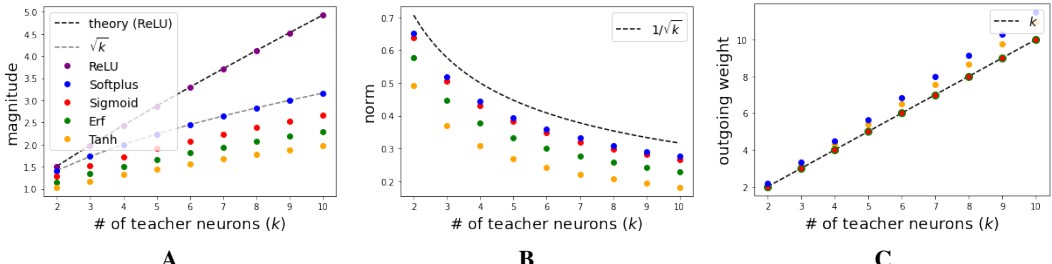

Figure 5: *Structure of the optimal solution of the one-neuron network for various activation functions.* We trained 20 seeds of one-neuron students learning from the unit-orthonormal teacher networks with $k = 2, ..., 10$ neurons. All students converge to the same optimal solution up to symmetries (that is, positive-scaling symmetry for ReLU and sign symmetry for odd activation functions such as tanh and erf). **A** For ReLU, the magnitude $\|w^*\|\|a^*\|$ exactly matches with the result of Corollary G.5. For softplus, the magnitude is very close to $\sqrt{k}$; for sigmoid, tanh, and erf, it is below $\sqrt{k}$. **B** The norm of the incoming vector is smaller than $1/\sqrt{k}$ for softplus, sigmoid, tanh, and erf. **C** The outgoing weight is larger than $k$ for softplus and tanh, and it is virtually $k$ for sigmoid and erf.

## C  Further Experiments

All experiments in this paper are implemented using the gradient flow package implemented by Brea et al. [16] which is particularly suited to studying gradient flow on the population loss. For activation functions for which there is an analytic formula, it is already implemented in the package; for the others, we used the approximator option for a speed-up compared to the numerical integration option. This method uses a neural network in the background fitted to approximate Gaussian integrals. We trained for $10^5$ ode iterations for erf and relu experiments; $10^3$ ode iterations for softplus, tanh, and sigmoid. For erf experiments, all seeds converged to configurations with gradient norm below $5 \cdot 10^{-8}$. For ReLU experiments, a fraction of seeds failed to converge (large gradient norm at the end of training). In Appendix C.3, we report among the seeds that succeeded in converging. Weights initialized as Gaussians with zero mean and standard deviation 0.1 or with Glorot initialization [43]. This is in contrast with Saad and Solla [2], Tian [5] where (order) parameters are initialized with positive values (as opposed to the rotationally symmetric initializations done in practice). For each $(n, k)$ pair, we implemented 10 or 20 seeds of random initializations.

### C.1  One-Neuron Network

Empirically, gradient flow converges to the point where all student-teacher correlations are $\frac{1}{k}$[4].

We know from Theorem 5.1 that at the non-trivial critical point all correlations are equal at correlation $1/\sqrt{k}$ and there is possibly another critical point at correlation $-1/\sqrt{k}$. Depending on the activation function, the point where correlations are $-1/\sqrt{k}$ might be

- either an equivalent of the optimum solution (for odd activation functions),
- or a saddle (for ReLU),
- or does not exist (for softplus).

The details can be found in the proofs for individual cases.

### C.2  Erf Experiments

In this section, we first numerically investigate whether the CA-optimal critical point is a saddle or minimum in Figure 6. Surprisingly, we find that the point turns from a saddle point to a minimum

---

[4]For odd activation functions, there are two solutions that are sign-symmetric: the first one where all correlations are $\frac{1}{\sqrt{k}}$ and its equivalent where all correlations are $-\frac{1}{\sqrt{k}}$. The first solution is plotted in Fig. 5 for a fine comparison on the positive scale.

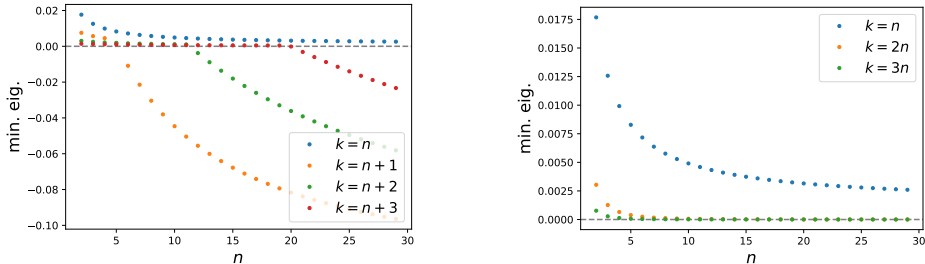

Figure 6: *The minimum eigenvalue of the Hessian at an optimal-CA point.* We numerically investigate whether a CA-optimal critical point is a strict saddle (min. eig. of the Hessian is negative) or a minimum (min. eig. of the Hessian is non-negative). Interestingly, the minimum eigenvalue turns from positive to negative as $n$ grows for $k = n + h$ for fixed $h = 1, 2, 3$ (left panel). Therefore in this regime, the CA-optimal cannot be the optimal solution of the non-convex problem for large $n$. For $k \gg n$, for example for $k = n, 2n, 3n$ (right panel), the min. eigenvalue is positive and it approaches zero as $n$ increases.

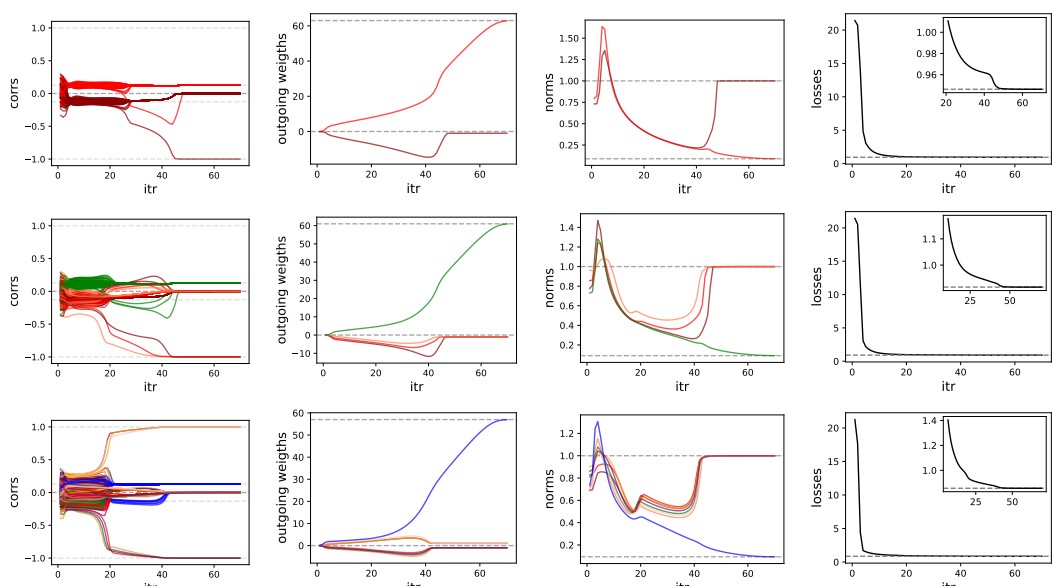

Figure 7: *Evolution of order parameters during convergence to a $(n-1)$-C-1-A solution: top $n = 2$, middle $n = 4$, bottom $n = 8$; and $k = 64$; representative seeds.* We can distinguish 3 phases: before iteration 20, after iteration 20, and beyond iteration 40. We observe that in the first phase of training (less than 5 iterations), the student neurons do not specialize into teacher neurons but approach the one-neuron solution. For $n = 2$ (top row), in the second phase, we observe that the first neuron implements an average of teacher neurons and the second neuron implements a copy of the remaining teacher neuron. In the second phase, in general, $n - 1$ neurons specialize to match one teacher neuron each (or its negative equivalent) and the $n$-th neuron splits its correlations into two groups: those that correspond to the teacher neurons being matched become negative and the others collapse on each other. Finally, in the third phase, the negative correlations converge to zero correlation, decoupling the student neurons from each other. All student neuron correlation signs can be flipped as long as the corresponding outgoing weight signs are flipped since the erf activation is odd. These examples illustrate the green regime (see Fig. 4 in the main text).

point in some regimes of $(n, k)$, despite that the configuration has the same structure of copying $n-1$ teacher neurons and taking an average of the remaining teacher neurons. When $k - n$ is fixed, and for large $n$, Figure 6 explains why gradient flow does not converge to the CA-optimal point.

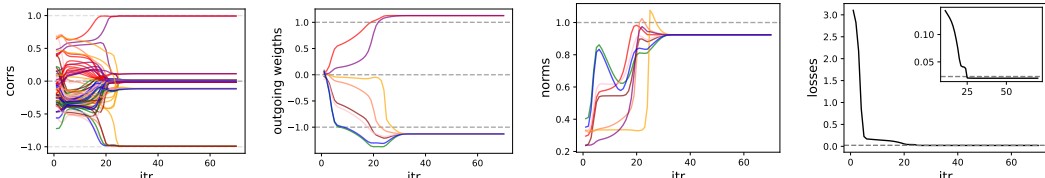

Figure 8: *Evolution of order parameters; P-n-C solution: for $n = 8$ and $k = 9$.* We observe that all student neurons match one teacher neuron in this case, however not perfectly at the end of training. This is an example of the red regime (see Fig. 4 in the main text), where the students converge to a perturbation of the $n$-copy configuration.

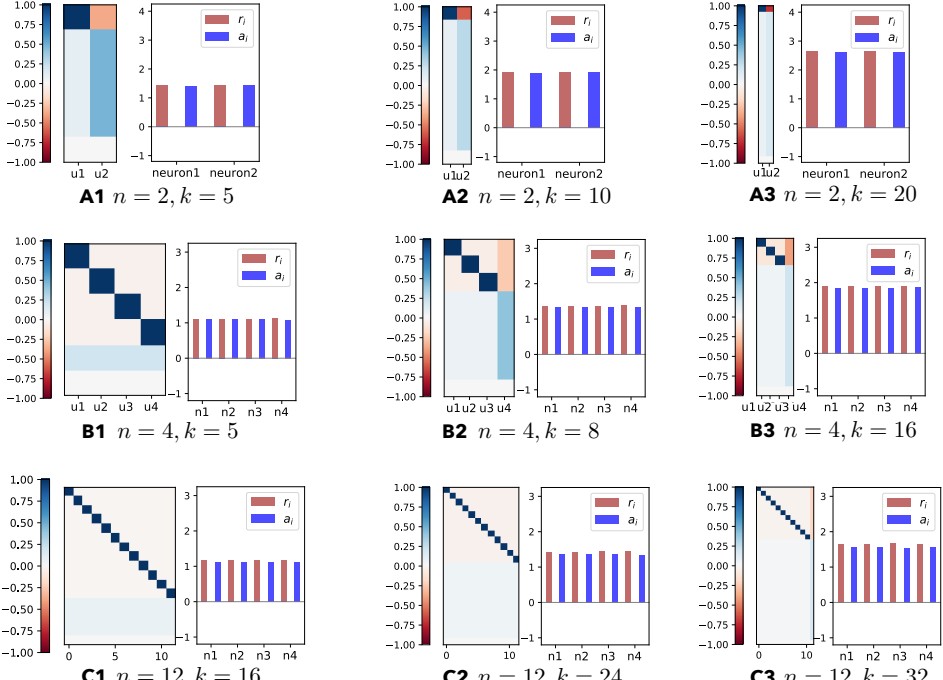

Figure 9: *Optimal configuration found by gradient flow for ReLU activation function for $n = 2$ (top); $n = 4$ (middle); $n = 12$ (bottom).* The last row of the correlation matrices represents $1 - \|u_i\|$ which is zero for all student neurons in all cases. *Top row, $n = 2$;* the optimal point is composed of a copy and an average neuron: the copy neuron is very close to one of the teacher neurons, and the average neuron is close to the average of the remaining teacher neurons while negatively correlating with the copied teacher neuron. The negative correlation increases in magnitude as $k$ increases. *Middle row, $n = 4$;* for $k = 5$, the optimal point is a perturbation of the all-copy configuration; for $k = 8, 16$, it is close to the $(n - 1)$-copy-1-average configuration. *Bottom row, $n = 12$;* for $k = 16, 24$, the optimal point is a perturbation of the all-copy configuration; for $k = 32$, it is close to the $(n - 1)$-copy-1-average configuration. In all regimes, the norms and outgoing weights of all student neurons are close to each other (for the bottom row only the first four neurons are shown).

We show some representative trajectories of gradient flow, in the regime when the CA-optimal critical point is a minimum in Figure 7 and in the regime when it is a saddle point in Figure 8.

## C.3   ReLU Experiments

In this subsection, we will present the structure of the minimum loss configuration found by gradient flow. In the regime $n \ll k$, the minimum loss configuration is qualitatively similar to the optimal-CA solution but without perfect decoupling. For $k$ that is slightly bigger than $n$, the gradient flow finds an "all-copy" configuration for $n = 4, 12$. The overall trend of the minimum loss configuration is

similar to the case of erf activation; however, as we do not have the analytic formula of correlations, a theoretical prediction for the optimal solution is left for future work.

## D  Constrained Optimization Formulation

Expressing the loss function in terms of the order parameters yields the 'projected' loss function

$$
L_{\text{proj}}^{n,k} = \sum_{i=1}^{n} \underbrace{a_i^2 g_\sigma(r_i, r_i, 1)}_{\text{student magnitude term}} + 2\sum_{i \neq i'} \underbrace{a_i a_{i'} g_\sigma(r_i, r_{i'}, \rho_{ii'})}_{\text{student-student interaction}} - 2\sum_{i=1}^{n}\sum_{j=1}^{k} \underbrace{a_i b_j g_\sigma(r_i, \|v_j\|, u_{ij})}_{\text{student-teacher interaction}} + C
\tag{25}
$$

where the constant term $C = \mathbb{E}_{x \sim \mathcal{D}}[f^*(x)^2]$. $L_{\text{proj}}$ has $n(k+2) + \binom{n}{2}$ parameters, that is $n$ output weights and $n(k+1) + \binom{n}{2}$ order parameters, instead of $n(d+1)$ parameters of the original loss function. For $d \gg k + 1 + \frac{n-1}{2}$, $L_{\text{proj}}$ has significantly less number of parameters.

In the special case $d = k$, the incoming vectors can be expressed as a linear combination of the teacher's incoming vectors, hence the correlations between them are not free (see Appendix D.1).

Each normalized incoming vector can be expressed as a sum of its projection on the span of the teacher's incoming vectors and an orthogonal component

$$
\frac{w_i}{r_i} = \sum_{j=1}^{k} u_{ij} v_j + v_i^\perp, \quad \sum_{j=1}^{k} u_{ij}^2 \leq 1,
\tag{26}
$$

and the inequality constraint pops up since $\|u_i\|^2 = 1 - \|v_i^\perp\|^2$ where $u_i = (u_{i1}, \ldots, u_{ik})$. For the correlations between the incoming vectors, we get

$$
\rho_{ii'} = u_i \cdot u_{i'} + v_i^\perp \cdot v_{i'}^\perp,
\tag{27}
$$

which yields the second set of constraints on the optimization problem

$$
\left| \rho_{ii'} - u_i \cdot u_{i'} \right| \leq \sqrt{1 - \|u_i\|^2}\sqrt{1 - \|u_{i'}\|^2} \quad \forall i' \neq i \in [n],
\tag{28}
$$

since we have $|v_i^\perp \cdot v_{i'}^\perp| \leq \|v_i^\perp\|\|v_{i'}^\perp\|$ due to the Cauchy-Schwarz inequality.

We note if all incoming vectors are in the span of the teacher's incoming vectors, we have that $\|u_i\| = 1$. As a result, the second set of inequalities in Eq. 28 collapse onto equalities, hence the secondary constraints are in fact equality constraints (see Appendix Eq. 34). We show that this is indeed the case for the non-trivial critical points of the one-neuron network in Section G.

In general, the constrained optimization formulation is possible for non-orthogonal teacher networks.

**Remark D.1.** *(Non-orthogonal teacher network) We can relax the assumption of orthogonality between $v_1, \ldots, v_k$ to linear independence. Let us collect the incoming vectors into a matrix $V = [v_1, \ldots, v_k] \in \mathbb{R}^{d \times k}$. The expansion in Eq. 26 can be rewritten as*

$$
\frac{w_i}{r_i} = \sum_{j=1}^{k} \gamma_{ij} v_j + v_i^\perp = V\Gamma_i + v_i^\perp
\tag{29}
$$

*where $\Gamma_i = (\gamma_{i1}, \ldots, \gamma_{ik}) \in \mathbb{R}^k$ and $v_i^\perp \cdot v_j = 0$ for all $j \in [k]$. The normalized vector has a unit norm, hence we have*

$$
\|V\Gamma_i + v_i^\perp\|^2 = \Gamma_i^T V^T V \Gamma_i + \|v_i^\perp\|^2 = 1.
\tag{30}
$$

*The correlation vector can be written as $u_i = V^T V \Gamma_i \in \mathbb{R}^k$ which yields the following constraint*

$$
u_i^T (V^T V)^{-1} u_i \leq 1.
\tag{31}
$$

### D.1  Equality Constraints in the Case $d = k$

In this case, the teacher incoming vectors $v_1, \ldots, v_k$ span the input domain $\mathbb{R}^d$. Each incoming vector (of the student) can be expressed as a linear combination of the teacher's incoming vectors

$$\frac{w_i}{r_i} = \sum_{j=1}^{k} u_{ij} v_j, \quad \sum_{j=1}^{k} u_{ij}^2 = 1, \tag{32}$$

and the equality constraint pops up since the normalized vector has a unit $\ell_2$ norm. The correlations between the incoming vectors are then expressed in terms of the student-teacher correlations

$$\rho_{ii'} = u_i \cdot u_{i'}. \tag{33}$$

Therefore the optimization problem is equivalent to

$$\min \quad \sum_{i=1}^{n} a_i^2 g_\sigma(r_i, r_i, 1) + 2 \sum_{i \neq i'} a_i a_{i'} g_\sigma\left(r_i, r_{i'}, \sum_{j=1}^{k} u_{ij} u_{i'j}\right) - 2 \sum_{i=1}^{n} \sum_{j=1}^{k} a_i b_j g_\sigma(r_i, \|v_j\|, u_{ij})$$

$$\text{subject to} \quad \sum_{j=1}^{k} u_{ij}^2 = 1, \ \ r_i \geq 0, \quad \text{for all } i \in [n]. \tag{34}$$

Since $\binom{n}{2}$ student-student correlations terms are not free, the problem has only $n(k + 2)$ free parameters and $k$ equality constraints, yielding $n(k + 1)$ effective parameters, which is the same number as the number of parameters of the original problem in the weight-space.

### D.2  Binary-Equality Constraints in the Case $d = k + 1$

In this case, there is only one direction orthogonal to the span of the teacher's incoming vectors (i.e. $v_i^\perp \parallel v_{i'}^\perp$). Therefore the general inequality constraint on $\rho_{ii'}$ reduces to

$$\rho_{ii'} = u_i \cdot u_{i'} \pm \sqrt{1 - \|u_i\|^2} \sqrt{1 - \|u_{i'}\|^2}. \tag{35}$$

### D.3  Three-Neuron Network

We present a case for the three-neuron network where the optimal solution of the constrained optimization problem may not be projected back to the weight space. In particular, let us consider a positive and monotonic activation function, i.e. $\sigma(x) > 0$. This implies that $g_\sigma > 0$ and that $g_\sigma$ is increasing in correlation. It is natural to expect that all outgoing weights are positive since this would bring the network function closer to the target function. If the network is overparameterized, some outgoing weights may be zero or even negative (balanced by a positive outgoing weight corresponding to the same incoming vector). Let us pick three positive outgoing weights $a_1, a_2, a_3 > 0$ and three corresponding student-student interaction terms

$$\text{minimize} \ \ a_1 a_2 g_\sigma(r_1, r_2, \rho_{12}) + a_1 a_3 g_\sigma(r_1, r_3, \rho_{13}) + a_2 a_3 g_\sigma(r_2, r_3, \rho_{23}) + ...,$$

$$\text{subject to} \ \ |\rho_{ii'} - u_i \cdot u_{i'}| \leq \sqrt{1 - \|u_i\|^2} \sqrt{1 - \|u_{i'}\|^2}. \tag{36}$$

Note that each $\rho_{ii'}$ is decoupled from each other. Since $g_\sigma$ is increasing in correlation, the minimum of each term above is achieved when

$$\rho_{ii'} = u_i \cdot u_{i'} - \sqrt{1 - \|u_i\|^2} \sqrt{1 - \|u_{i'}\|^2}. \tag{37}$$

This implies that the inequality is tight and therefore $v_i^\perp \parallel v_{i'}^\perp$ moreover,

$$v_i^\perp = \sqrt{1 - \|u_i\|^2} v^\perp \quad \text{and} \quad v_{i'}^\perp = -\sqrt{1 - \|u_{i'}\|^2} v^\perp \tag{38}$$

up to a sign flip. However, it is not possible that the three vectors all have pairwise flipped directions to each other as we would have $(-) \cdot (-) = (+)$. If the optimal solution of the constrained optimization problem verifies $\|u_i\|^2 < 1$, then we conclude that it cannot be mapped back to the weight space; in this case, the optimal $L_{\text{proj}}$ would only give a lower bound on the optimal loss of the weight-space.

# E  Copy-Average Critical Points

**Theorem E.1.** *Assume that $\sigma$ is the erf activation function. We pick a copy-average parameter vector*

$$\theta^* = (w_1^*, a_1^*) \oplus ... \oplus (w_n^*, a_n^*) \tag{39}$$

*where $(w_i^*, a_i^*)$ is a non-trivial critical point when learning from a unit-orthonormal teacher $f_i^*$ with the incoming vectors $v_{s_{i-1}+1}, ..., v_{s_i}$ shown in Eq. 11. Then $\theta^*$ is a critical point of the loss function $L^{n,k}$ where the target function is $f^*(x) = \sum_{j=1}^k \sigma(v_j \cdot x)$.*

*Proof.* Let us write down the partial derivatives with respect to the outgoing weights and incoming vectors

$$\frac{d}{da_i} L^{n,k}(\theta^*) = 2\mathbb{E}_{x\sim\mathcal{D}}[\sigma(w_i^* \cdot x)(\sum_{j=1}^n a_j^*\sigma(w_j^* \cdot x) - f^*(x))],$$

$$\frac{d}{dw_i} L^{n,k}(\theta^*) = 2a_i^*\mathbb{E}_{x\sim\mathcal{D}}[\sigma'(w_i^* \cdot x)x(\sum_{j=1}^n a_j^*\sigma(w_j^* \cdot x) - f^*(x))]. \tag{40}$$

We will show that they are equivalent to the following

$$\frac{d}{da_i} L^{n,k}(\theta^*) = 2\mathbb{E}_{x\sim\mathcal{D}}[\sigma(w_i^* \cdot x)(a_i^*\sigma(w_i^* \cdot x) - f_i^*(x))],$$

$$\frac{d}{dw_i} L^{n,k}(\theta^*) = 2a_i^*\mathbb{E}_{x\sim\mathcal{D}}[\sigma'(w_i^* \cdot x)x(a_i^*\sigma(w_i^* \cdot x) - f_i^*(x))], \tag{41}$$

which implies that the partial derivatives are zero, since $(w_i^*, a_i^*)$ is a critical point of the loss

$$L^{1,\ell_i} = \mathbb{E}_{x\sim\mathcal{D}}[(a\sigma(w \cdot x) - f_i^*(x))^2]. \tag{42}$$

Since $(w_i^*, a_i^*)$ is the optimal solution of the teacher network generated by $v_{s_{i-1}+1}, ..., v_{s_i}$, from Theorem 5.1, we have that $w_i^*$ is in the span of $v_{s_{i-1}+1}, ..., v_{s_i}$. We have that

$$w_i^* \cdot w_{i'}^* = 0 \quad \text{and} \quad w_i^* \cdot v_j = 0 \quad \text{for} \quad j \in [k] \setminus [s_{i-1}+1, s_i] \tag{43}$$

since the two incoming vectors are in the span of two orthogonal subspaces respectively and $w_i^*$ is orthogonal to all other teacher incoming vectors that are outside of the span of $v_{s_{i-1}+1}, ..., v_{s_i}$. For two orthogonal vectors say $w_i^*$ and $v$, we have that

$$\mathbb{E}_{x\sim\mathcal{D}}[\sigma_1(w_i^* \cdot x)\sigma_2(v \cdot x)] = \mathbb{E}_{x\sim\mathcal{D}}[\sigma_1(w_i^* \cdot x)]\mathbb{E}_{x\sim\mathcal{D}}[\sigma_2(v \cdot x)] \tag{44}$$

which is zero if at least one of $\sigma_1$ and $\sigma_2$ is odd. This implies the first equation in 41 for the partial derivatives with respect to the outgoing weights. In order to show the second equation for the partial derivatives with respect to the incoming vectors, let us define $S_j = [s_{j-1}+1, s_j]$,

$$W_{i,j} = \mathbb{E}_{x\sim\mathcal{D}}[\sigma'(w_i^* \cdot x)x(a_j^*\sigma(w_j^* \cdot x) - \sum_{k\in S_j}\sigma(v_k \cdot x))], \tag{45}$$

and note that it suffices to show that $W_{i,j} = 0$ for all $i \neq j$. This is true if and only if $W_{i,j} \cdot \bar{v}_\ell = 0$ where $\{\bar{v}_1, ..., \bar{v}_d\}$ form an orthogonal basis of $\mathbb{R}^d$. Let us choose $\bar{v}_1 = v_1, ..., \bar{v}_k = v_k$ and that $\bar{v}_{k+1}, ..., \bar{v}_d$ completes the basis if $d > k$. One can observe that $W_{i,j} \cdot \bar{v}_\ell = 0$ for $k+1 \leq \ell \leq d$ since $x \cdot \bar{v}_\ell$ is an independent Gaussian from the others. Hence, the expectation, i.e. $W_{i,j} \cdot \bar{v}_\ell$, factorizes with a factor of $\mathbb{E}[x \cdot \bar{v}_\ell]$ which is zero.

It remains to check $W_{i,j} \cdot v_l = 0$ for all $v_l$'s. We split the analysis into two cases. If $v_l \notin S_j$, then $x \cdot v_l$ is independent from $x \cdot w_j^*$ and from $x \cdot v_k$ for $k \in S_j$. Hence $W_{i,j}$ splits into

$$W_{i,j} \cdot v_l = \mathbb{E}_{x\sim\mathcal{D}}[\sigma'(w_i^* \cdot x)x \cdot v_l]\mathbb{E}_{x\sim\mathcal{D}}[(a_j^*\sigma(w_j^* \cdot x) - \sum_{k\in S_j}\sigma(v_k \cdot x))] = 0, \tag{46}$$

where the second term in the product is zero because $w_j^* \cdot x$ and $v_k \cdot x$ are centered Gaussian and $\sigma$ is odd. For the second case, where $v_l \in S_j$, using the fact that $x \cdot v_l$ is independent from $w_i^* \cdot x$ and from $x \cdot v_k$ for $l \neq k$, we have

$$W_{i,j} \cdot v_l = \mathbb{E}_{x \sim \mathcal{D}}[\sigma'(w_i^* \cdot x)]\mathbb{E}_{x \sim \mathcal{D}}[x \cdot v_l(a_j^* \sigma(w_j^* \cdot x) - \sum_{k \in S_j} \sigma(v_k \cdot x))] \tag{47}$$

$$= \mathbb{E}_{x \sim \mathcal{D}}[\sigma'(w_i^* \cdot x)]\mathbb{E}_{x \sim \mathcal{D}}[x \cdot v_l(a_j^* \sigma(w_j^* \cdot x) - \sigma(v_l \cdot x))] \tag{48}$$

$$= \mathbb{E}_{x \sim \mathcal{D}}[\sigma'(w_i^* \cdot x)]\Big(\mathbb{E}_{x \sim \mathcal{D}}[a_j^* \sigma(w_j^* \cdot x)x \cdot v_l] - \mathbb{E}_{x \sim \mathcal{D}}[\sigma(v_l \cdot x)x \cdot v_l]\Big). \tag{49}$$

Applying Stein's Lemma to both terms on the right we have

$$W_{i,j} \cdot v_l = \mathbb{E}_{x \sim \mathcal{D}}[\sigma'(w_i^* \cdot x)]\Big(\mathbb{E}[a_j^* r_j^* u^* \sigma'(r_j^* Z)] - \mathbb{E}[\sigma'(Z)]\Big), \tag{50}$$

where $Z$ is standard Gaussian and $r_j^* = \sqrt{\frac{1}{2k-1}}$, $u^* = \sqrt{\frac{1}{k}}$, $a_j^* = k$ (parameters of erf). Hence we want to show that

$$a_j^* r_j^* u^* \mathbb{E}[\sigma'(r_j^* Z)] = \mathbb{E}[\sigma'(Z)]. \tag{51}$$

To show this, we use the following relation [2, 3]

$$g_{\text{erf}}(r, r, u) = \mathbb{E}[\sigma(rx)\sigma(ry)] = \frac{2}{\pi} \arcsin\left(\frac{r^2 u}{r^2 + 1}\right). \tag{52}$$

Differentiating with respect to the correlation $u$ we have

$$\frac{d}{du} g_{\text{erf}}(r, r, u) = r^2 \mathbb{E}[\sigma'(rx)\sigma'(ry)] = \frac{2}{\pi} \frac{1}{\sqrt{1 - u^2 \left(\frac{r^2}{r^2+1}\right)^2}} \frac{r^2}{(r^2 + 1)}. \tag{53}$$

In particular, at correlation zero, we get

$$\mathbb{E}[\sigma'(rx)] = \sqrt{\frac{2}{\pi} \frac{1}{(r^2 + 1)}} \tag{54}$$

Therefore, we have

$$\mathbb{E}[\sigma'(r_j^* x)] = \sqrt{\frac{2}{\pi}\left(\frac{2k - 1}{2k}\right)}, \quad \mathbb{E}[\sigma'(x)] = \sqrt{\frac{1}{\pi}}, \tag{55}$$

which implies 51 and the proof is complete. □

For general activation functions, using the substitution in Eq.41, the first partial derivatives in Eq. 40 reduce to

$$\frac{d}{da_i} L^{n,k}(\theta^*) = 2\sum_{i \neq i'} a_{i'}^* g_\sigma(\|w_i^*\|, \|w_{i'}^*\|, 0) - 2(k - \ell_i)g_\sigma(\|w_i^*\|, 1, 0) \tag{56}$$

which is in general non-zero if $\sigma$ is not odd.

**Lemma E.2.** *Assume $\ell_2 > \ell_1 \geq 1$. We have that*

$$L_{erf}^*(\ell_2 + 1) - L_{erf}^*(\ell_2) < L_{erf}^*(\ell_1 + 1) - L_{erf}^*(\ell_1). \tag{57}$$

*Proof.* We first show that the function $x^2 \arcsin(\frac{1}{2x})$ is increasing for $x \geq 1$ and convex for $x > 0$. Using the Taylor expansion of arcsin, we have that

$$f(x) = x^2 \arcsin\left(\frac{1}{2x}\right) = \frac{x}{2} + \frac{1}{2 \cdot 3}\frac{1}{2^3 x} + \frac{1 \cdot 3}{2 \cdot 4 \cdot 5}\frac{1}{2^5 x^3} + \dots \tag{58}$$

where the higher-order terms all have positive coefficients. The first derivative is

$$f'(x) = \frac{1}{2} - \frac{1}{2 \cdot 3}\frac{1}{2^3 x^2} - \frac{1 \cdot 3 \cdot 3}{2 \cdot 4 \cdot 5}\frac{1}{2^5 x^4} + \dots \tag{59}$$

which is positive for $x \geq 1$ since we have

$$\frac{1}{2 \cdot 3} \frac{1}{2^3 x^2} + \frac{1 \cdot 3 \cdot 3}{2 \cdot 4 \cdot 5} \frac{1}{2^5 x^4} + ... < \frac{1}{2^2 x^2} + \frac{1}{2^4 x^4} + ... \leq \frac{1}{4} + \frac{1}{4^2} + \frac{1}{4^3} + ... = \frac{1}{3}. \quad (60)$$

The second derivative is

$$f''(x) = 2\frac{1}{2 \cdot 3} \frac{1}{2^3 x^3} + 4\frac{1 \cdot 3 \cdot 3}{2 \cdot 4 \cdot 5} \frac{1}{2^5 x^3} + ... \quad (61)$$

which is positive for positive $x$.

First, let us show that Eq. 18 holds. Plugging in the analytic expressions for $L^*_{\text{erf}}(0, \cdot)$ and $L^*_{\text{erf}}(\cdot)$, it is equivalent to

$$\ell_0 \frac{2}{\pi} \arcsin\left(\frac{1}{2}\right) > \ell_0 \frac{2}{\pi} \arcsin\left(\frac{1}{2}\right) - \frac{2}{\pi}\left((\ell_1 + \ell_0)^2 \arcsin\left(\frac{1}{2(\ell_1 + \ell_0)}\right) - \ell_1^2 \arcsin\left(\frac{1}{2\ell_1}\right)\right). \quad (62)$$

Since $f(x)$ is increasing for $x \geq 1$, the second term inside the parenthesis is positive, hence the inequality holds.

We will now prove the statement of the Lemma. It suffices to show the following for all $\ell = \ell_2 \geq 2$

$$L^*_{\text{erf}}(\ell + 1) - L^*_{\text{erf}}(\ell) < L^*_{\text{erf}}(\ell) - L^*_{\text{erf}}(\ell - 1) \quad (63)$$

since then we can continue to decrease $\ell$ by one, i.e. $\ell - 1, \ell - 2, ...$, until we reach $\ell_1$. Eq. 63 is equivalent to

$$L^*_{\text{erf}}(\ell + 1) - 2L^*_{\text{erf}}(\ell) + L^*_{\text{erf}}(\ell - 1) < 0, \quad (64)$$

that is the second-order finite difference, similar to the second-derivative of a continuous function. The proof is completed by observing that $L^*_{\text{erf}}(\cdot)$ is a discrete-concave function since its continuous interpolation

$$L^*_{\text{erf}}(x) = x \arcsin\left(\frac{1}{2}\right) - x^2 \arcsin\left(\frac{1}{2x}\right) \quad (65)$$

is concave for $x > 0$ since it can be written as $L^*_{\text{erf}}(x) = \alpha x - f(x)$ where $f$ is convex for $x > 0$. $\quad \square$

Lemma E.2 tells us that if we add one neuron to the teacher, then it is better to approximate it by the student neuron that already approximates many teacher neurons. Applying Lemma E.2 iteratively, we get

$$L^*_{\text{erf}}(k-1) + L^*_{\text{erf}}(1) < L^*_{\text{erf}}(k-2) + L^*_{\text{erf}}(2) < ... < L^*_{\text{erf}}(\ell_2+1) + L^*_{\text{erf}}(\ell_1) < L^*_{\text{erf}}(\ell_2) + L^*_{\text{erf}}(\ell_1+1)$$

where $\ell_2 = \frac{k}{2}, \ell_1 = \frac{k}{2} - 1$ if $k$ is even and $\ell_2 = \frac{k+1}{2}, \ell_1 = \frac{k-3}{2}$ if $k$ is odd.

**Theorem E.3.** *Consider a unit-orthonormal teacher network $f^*(x) = \sum_{j=1}^k \sigma(v_j \cdot x)$ and the erf activation function. For an under-parameterized student network with $n$ neurons, the minimum-loss copy-average configuration up to permutations (of the student and teacher neurons) is*

$$\theta = (\epsilon_1 v_1, \epsilon_1) \oplus ... \oplus (\epsilon_{n-1} v_{n-1}, \epsilon_{n-1}) \oplus (\epsilon_n w_n^*, \epsilon_n a_n^*) \quad (66)$$

*where $\epsilon_i \in \{\pm 1\}$ and $(w_n^*, a_n^*)$ is given by Corollary 5.2 after substituting $k$ with $k-n+1$.*

*Proof.* We will conclude with a simple argument that the minimum-loss CA configuration for a multi-neuron network with $n$ neurons is $(n-1)$-C-1-A. In particular, if there are two averaging neurons inside the student network, we can redistribute the teacher neurons shared between these two to a lower-loss CA configuration by ensuring that one student neuron copies and the other student neuron averages (see Lemma E.2). The minimum-loss CA point is then achieved among CA configurations where at least $n-1$ neurons each copy a single teacher neuron (of the $k$ possible ones). The remaining student neuron can be treated as a single-neuron network learning from a teacher with $k-n+1$ neurons – for which we know the optimal solution is to average (Theorem 5.1). $\quad \square$

### E.1 Number of CA Critical Points

There is a combinatorial number of $(\ell_1, ..., \ell_n)$-CAC critical points, that is

$$c(\ell_1, ..., \ell_n)\binom{k}{\ell_1}...\binom{k - (\ell_1 + ... + \ell_n)}{\ell_n} \tag{67}$$

where $c_n := c(\ell_1, ..., \ell_n)$ counts distinguishable permutations between the neurons of the student network, and the binomial coefficients stand for grouping teacher neurons into $n$ non-empty buckets.

If $\ell_1 = ... = \ell_n$, permutation between the student neurons is already counted when distributing the teacher neurons, hence $c_n = 1$. If all $\ell_1, ..., \ell_n$ are distinct from each other, we have that $c_n = n!$ since we swap all pairs of student neurons after assigning groups of teacher neurons. In general, let $c_i$ denote the number of $i$'s among $\ell_1, ..., \ell_n$ for all $i = 1, .., k$; the formula for the permutation-factor is given by

$$c(\ell_1, ..., \ell_n) = \frac{n!}{c_1!...c_k!}. \tag{68}$$

## F  General Properties of the Interaction Function

In this Section, we introduce some general properties of the interactions. We use these only for the one-neuron network in this paper (see Section G), however, these properties are likely to play a role in studying the networks with two or more neurons.

We first present the partial derivative of a general interaction function, i.e. two activation functions may be different, for example, if the student activation function does not match the teacher, with respect to the correlation in a simple expression in Lemma F.1. In the second part, we present a property of the activation function sufficient for Assumption 3.1 (ii), and show that the differentiable activation functions studied in this paper satisfy this property in Lemma F.2.

**Lemma F.1.** *Assume that functions $\sigma_1$ and $\sigma_2$ are differentiable. The partial derivative of the following Gaussian integral term $\mathbb{E}[\sigma_1(r_1 x)\sigma_2(r_2 y)]$ with respect to the correlation $\mathbb{E}[xy] = u$ is*

$$\frac{d}{du}\mathbb{E}[\sigma_1(r_1 x)\sigma_2(r_2 y)] = r_1 r_2 \mathbb{E}[\sigma_1'(r_1 x)\sigma_2'(r_2 y)]. \tag{69}$$

We apply the Lemma for $\sigma_1 = \sigma_2 = \sigma$ in the main text in Eq. 9.

*Proof.* We compute the derivative of $\mathbb{E}[\sigma_1(r_1 x)\sigma_2(r_2 y)]$ by making the correlation $u$ explicit. Denote $u' = \sqrt{1 - u^2}$ and $y = ux + u'z$. After the computation, we use Stein's lemma to reach the desired formula.

$$\partial_u \mathbb{E}[\sigma_1(r_1 x)\sigma_2(r_2 y)] = r_2 \mathbb{E}[\sigma_1(r_1 x)\sigma_2'(r_2 y)x] - \frac{r_2 u}{u'}\mathbb{E}[\sigma_1(r_1 x)\sigma_2'(r_2 y)z] \tag{70}$$

where $x$ and $z$ are independent standard Gaussians. Here is a reminder for Stein's Lemma for a standard Gaussian $z$

$$\mathbb{E}[v(z)z] = \mathbb{E}[v'(z)]. \tag{71}$$

To remove $x$ in the first term, we apply Stein's formula for $v(x) = \sigma_1(r_1 x)\sigma_2'(r_2(ux + u'z))$ yielding

$$r_1 r_2 \mathbb{E}[\sigma_1'(r_1 x)\sigma_2'(r_2 y)] + r_2^2 u\mathbb{E}[\sigma_1(r_1 x)\sigma_2''(r_2 y)]. \tag{72}$$

To remove $z$ in the second term, we apply Stein's formula for $v(z) = \sigma_2'(r_2(ux + u'z))$ by considering fixed $x$ which yields

$$-r_2^2 u\mathbb{E}[\sigma_1(r_1 x)\sigma_2''(r_2 y)]. \tag{73}$$

Summing up the two terms completes the proof. $\qquad\square$

For softplus that is increasing and convex, using Lemma F.1 for $\sigma_1 = \sigma_2 = \sigma$ twice, we infer that the interaction $g$ is also increasing and convex in $u$. Hence, for $u < 0$, Assumption 3.1 (ii) holds for softplus. However, for the other activation functions, using second-order derivatives does not suffice to show the assumption. We will propose a new property of the activation function that implies that the interaction satisfies Assumption 3.1 (ii) and prove that softplus with $\beta \le 2$, sigmoid, tanh, and erf satisfy this property.

**Lemma F.2.** *If the activation function $\sigma$ is thrice-differentiable and it satisfies*

$$\sigma'(x) - x\sigma''(x) + \sigma'''(x) > 0, \tag{74}$$

*then its interaction satisfies Assumption 3.1 (ii) for all $u \in (-1, 1)$. Softplus with $\beta \in (0, 2]$, sigmoid, tanh, and erf activation functions satisfy the above inequality.*

*Proof.* Let us first write out Assumption 3.1 (ii) explicitly using Lemma F.1

$$r_1 u \mathbb{E}[\bar{\sigma}'(r_1 x)\bar{\sigma}'(y)] < \mathbb{E}[\bar{\sigma}(r_1 x)\bar{\sigma}(y)]. \tag{75}$$

where $\bar{\sigma}(x) = \sigma'(x)$. Using Stein's Lemma for $v(x) = \bar{\sigma}(r_1 x)\bar{\sigma}'(y)$, we get

$$\mathbb{E}[\bar{\sigma}(r_1 x)\bar{\sigma}'(y)x] = \mathbb{E}[\bar{\sigma}'(r_1 x)\bar{\sigma}'(y)]r_1 + \mathbb{E}[\bar{\sigma}(r_1 x)\bar{\sigma}''(y)]u. \tag{76}$$

The desired inequality is equivalent to

$$\mathbb{E}[\bar{\sigma}(r_1 x)(\bar{\sigma}(y) - \bar{\sigma}'(y)xu + \bar{\sigma}''(y)u^2)] > 0. \tag{77}$$

Let us introduce $f(x) = \bar{\sigma}(x) - x\bar{\sigma}'(x) + \bar{\sigma}''(x)$. For $y = ux + u'z$ where $u' = \sqrt{1 - u^2}$, we have the conditional average of $y$ fixing $x$ (we drop conditioning on the right-hand terms for convenience)

$$\begin{aligned}
\mathbb{E}[f(y)|x] &= \mathbb{E}[\bar{\sigma}(y)] - \mathbb{E}[y\bar{\sigma}'(y)] + \mathbb{E}[\bar{\sigma}''(y)] \\
&= \mathbb{E}[\bar{\sigma}(y)] - ux\mathbb{E}[\bar{\sigma}'(y)] - \mathbb{E}[u'z\bar{\sigma}'(y)] + \mathbb{E}[\bar{\sigma}''(y)] \\
&= \mathbb{E}[\bar{\sigma}(y)] - ux\mathbb{E}[\bar{\sigma}'(y)] - (u')^2\mathbb{E}[\bar{\sigma}''(y)] + \mathbb{E}[\bar{\sigma}''(y)] \\
&= \mathbb{E}[\bar{\sigma}(y)] - ux\mathbb{E}[\bar{\sigma}'(y)] + u^2\mathbb{E}[\bar{\sigma}''(y)],
\end{aligned} \tag{78}$$

where second last equality comes from Stein's Lemma for $v(z) = \bar{\sigma}'(ux + u'z)$. Hence the desired inequality is equivalent to

$$\mathbb{E}[\bar{\sigma}(r_1 x)f(y)] > 0. \tag{79}$$

By straightforward calculus, we will show that $f(x) > 0$, or that $f(x) \geq 0$ and $f(x) = 0$ if and only if $x = 0$. In the latter case, the expectation in Eq. 79 is positive since $f(y) > 0$ for some $y$ values of the integrand. First, for the sigmoid and tanh activation functions, for which we have

$$\bar{\sigma}(x) = \frac{e^x}{(e^x + 1)^2}, \ \bar{\sigma}'(x) = \frac{e^x(1 - e^x)}{(e^x + 1)^3}, \ \bar{\sigma}''(x) = \frac{e^x(e^{2x} - 4e^x + 1)}{(e^x + 1)^4}. \tag{80}$$

Hence, we can explicitly write $f$ as

$$f(x) = \frac{e^x}{(e^x + 1)^2} - x\frac{e^x(1 - e^x)}{(e^x + 1)^3} + \frac{e^x(e^{2x} - 4e^x + 1)}{(e^x + 1)^4} \tag{81}$$

$$= \frac{e^x}{(e^x + 1)^4}((e^x + 1)^2 - x(1 - e^x)(e^x + 1) + (e^{2x} - 4e^x + 1)). \tag{82}$$

Therefore showing $f(x) > 0$ is equivalent to showing that the factor on the right, that is,

$$2e^x(e^x - 1) + 2 - x(1 - e^{2x}) \tag{83}$$

is positive. For $x < 0$, we have $e^x < 1$ which implies $-x(1 - e^{2x}) > 0$ and $(1 - e^x)e^x \leq 1/4$ due to the inequality of arithmetic and geometric means hence the first term is upper bounded by $-1/2$ and since we have $+2$, the whole term is positive. For $x \geq 0$, we have $e^x \geq 1$, hence we can rewrite the inequality as a sum of non-negative terms

$$2e^x(e^x - 1) + 2 + x(e^{2x} - 1) > 0. \tag{84}$$

Let us now handle the case of erf. Its first three derivatives are given by

$$\bar{\sigma}(x) = \frac{2}{\sqrt{\pi}}e^{-x^2/2}, \bar{\sigma}'(x) = -\frac{2}{\sqrt{\pi}}xe^{-x^2/2}, \bar{\sigma}''(x) = \frac{2}{\sqrt{\pi}}(x^2 e^{-x^2/2} - e^{-x^2/2}) \tag{85}$$

Hence, we can explicitly write $f$ as

$$f(x) = \frac{2}{\sqrt{\pi}}e^{-x^2/2}(1 + xx + x^2 - 1) = \frac{4}{\sqrt{\pi}}e^{-x^2/2}x^2 \tag{86}$$

that is non-negative for all $x$ and zero iff $x = 0$.

Finally, for the softplus activation function with $\beta \in (0, 2]$, we have the following derivatives

$$\bar{\sigma}(x) = \frac{e^{\beta x}}{(e^{\beta x} + 1)}, \ \bar{\sigma}'(x) = \frac{\beta e^{\beta x}}{(e^{\beta x} + 1)^2}, \ \bar{\sigma}''(x) = \frac{\beta^2 e^{\beta x}(1 - e^{\beta x})}{(e^{\beta x} + 1)^3}. \tag{87}$$

Plugging in the function $f$, we get

$$f(x) = \frac{e^{\beta x}}{(e^{\beta x} + 1)} - x\frac{\beta e^{\beta x}}{(e^{\beta x} + 1)^2} + \frac{\beta^2 e^{\beta x}(1 - e^{\beta x})}{(e^{\beta x} + 1)^3} \tag{88}$$

$$= \frac{e^{\beta x}}{(e^{\beta x} + 1)^3}((e^{\beta x} + 1)^2 - x\beta(e^{\beta x} + 1) + \beta^2(1 - e^{\beta x})) \tag{89}$$

Therefore showing $f(x) > 0$ is equivalent to showing that the factor on the right, that is,

$$e^{2\beta x} + e^{\beta x}(2 - x\beta - \beta^2) + 1 - x\beta + \beta^2 \tag{90}$$

is positive. For $x \leq 0$, we have that $-x\beta > 0$ and $2 - \beta^2 \geq -2$ since $\beta \leq 2$, hence it is sufficient to show that the following is positive

$$e^{2\beta x} - 2e^{\beta x} + 1 + \beta^2 = (e^{\beta x} - 1)^2 + \beta^2 \tag{91}$$

which is a sum of squares. For $x > 0$, in the rest of the proof we will show that

$$e^{\beta x}(e^{\beta x} + 2 - x\beta - \beta^2) + 1 - x\beta + \beta^2 > 0, \tag{92}$$

for $\beta \in (0, 2]$. Using $e^{\beta x} \geq (\beta x)^2/2 + \beta x + 1$, it suffices to show that

$$e^{\beta x}((\beta x)^2/2 + 3 - \beta^2) + 1 - x\beta + \beta^2 > 0. \tag{93}$$

If $(\beta x)^2/2 + 3 - \beta^2 \geq 1$, then the first term is bigger than $\beta x + 1$ hence the above term is positive. The remaining possibility is that we have

$$\frac{x^2}{2} < 1 - \frac{2}{\beta^2}. \tag{94}$$

$\beta \leq 2$ implies $x < 1$ and $x^2 > 0$ implies $\beta > \sqrt{2}$. Hence we have $-x\beta + \beta^2 > 0$ since $\beta > x$. Therefore, if we have $(\beta x)^2/2 + 3 - \beta^2 \geq 0$, Eq. 92 is positive. Assuming the opposite, we get

$$\frac{x^2}{2} < 1 - \frac{3}{\beta^2}, \tag{95}$$

$\beta \leq 2$ implies $x < 1/\sqrt{2}$ and $x^2 > 0$ implies $\beta > \sqrt{3}$.

Going back to Eq. 92, what remains to show is that it is positive in the domain $x < 1/\sqrt{2}$, $\beta \in (\sqrt{3}, 2]$. It suffices to show that $e^{\beta x} + 2 - x\beta - \beta^2 > 0$. Assuming the contrary implies $e^{\beta x} < x\beta + 2$ since $\beta \leq 2$. We can then deduce that $x\beta < c = 1.2$ since otherwise we would have

$$e^{\beta x} = 1 + \beta x + \frac{(\beta x)^2}{2!} + \frac{(\beta x)^3}{3!} + ... \tag{96}$$

$$\geq 1 + \beta x + \frac{c^2}{2!} + \frac{c^3}{3!} + ... = 1 + \beta x + (e^c - c - 1) > 1 + \beta x + 1 \tag{97}$$

which implies a contradiction. $c$ can be chosen smaller but this will be enough for our purposes.

Assuming $e^{\beta x} + 2 - x\beta - \beta^2 \leq 0$, let us expand Eq. 92

$$e^{\beta x}(e^{\beta x} + 2 - x\beta - \beta^2) + 1 - x\beta + \beta^2 \geq \ (\text{using } e^{\beta x} < \beta x + 2) \tag{98}$$

$$(x\beta + 2)e^{\beta x} + (x\beta + 2)(2 - x\beta - \beta^2) + 1 - x\beta + \beta^2 = \tag{99}$$

$$(x\beta + 2)e^{\beta x} - (x\beta)^2 - (1 + \beta^2)x\beta + 5 - \beta^2 > \ (\text{using } e^{\beta x} > \beta x + 1) \tag{100}$$

$$7 - \beta^2 + (2 - \beta^2)x\beta \geq 3 - 2x\beta > 0 \tag{101}$$

where in the last inequality we used $x\beta < 1.2$. We note that this inequality holds for slightly larger $\beta$ using the same technique, however, for significantly larger $\beta$, the property breaks down. $\qquad \square$

# G   The One-Neuron Network

We study the critical points of the following loss function

$$L^{1,k}(w,a) = \mathbb{E}_{x\sim\mathcal{D}}[a\sigma(w\cdot x) - \sum_{j=1}^{k} b_j\sigma(v_j\cdot x)], \tag{102}$$

in particular, the optimal solution. For $n=1$, all configurations of order parameters are realizable in the weight space, therefore, the optimal solution of the following loss (repeating Eq. 14)

$$L^{1,k}_{\text{proj}} = a^2 g_\sigma(r,r,1) - 2a\sum_{j=1}^{k} b_j g_\sigma(r,\|v_j\|,u_j) + \text{const}, \quad \text{subject to} \sum_{j=1}^{k} u_j^2 \leq 1, r \geq 0, \tag{103}$$

is equivalent to the optimal solution in the weight space. Let us denote the unit ball by $B = \{(u_1,...,u_k) \mid u_1^2 + ... + u_k^2 \leq 1\}$. Its interior is denoted by $\text{int } B$ and its boundary is denoted by $\partial B$.

We will present the results for the one-neuron network in five parts

1. In Subsection G.1, we give a proof of Proposition 4.1: any non-trivial critical point $(w,a)$ of $L^{1,k}$ satisfies that $w$ is in the span of the teacher's incoming vectors if the activation function satisfies Assumption 3.1 (i). Moreover, we show in Lemma G.2 that the corresponding order parameters should satisfy a Lagrangian condition (Eq. 105).

2. In Subsection G.2, we characterize the topology of the loss landscape in terms of its critical points for the activation functions studied in this paper and for the unit-orthonormal teacher. Our results for the one-neuron network are strong in the sense that it gives all possible critical points of the loss landscape.

   (a) In Subsection G.2.1, we give a proof of Theorem 5.1: for general activation functions satisfying Assumption 3.1, any non-trivial critical point of the one-neuron network attains equal correlations that are either $1/\sqrt{k}$ or $-1/\sqrt{k}$.

   (b) In Subsection G.2.2, we study the two-dimensional loss obtained after applying Theorem 5.1. From its derivative constraints, we get a fixed point equation (Eq. 120) that needs to be satisfied by the incoming vector norm $r$ at any non-trivial critical point with equal correlations $u$. Numerically, we show that there is a unique solution of the fixed point equation for $u > 0$ for differentiable activation functions studied in this paper (Fig. 10). Finally, we give some sufficient conditions in Eq. 121 to prove uniqueness based on log-concavity; numerically, these are shown to be satisfied by softplus and sigmoid activation functions.

3. In Subsection G.3, we give a proof of Corollary 5.2 by solving the two-dimensional loss for the erf activation function. Moreover, from the proof, we conclude that there are exactly two non-trivial critical points identical up to the mirror symmetry of erf (since it is odd); and these are the optimal solutions for the loss landscape.

4. In Subsection G.4, we present and prove Corollary G.5 by solving the two-dimensional loss for the ReLU activation function. We find that there are two non-trivial critical points of the loss function: a saddle 'point' at correlation $-1/\sqrt{k}$ and the optimal 'solution' at correlation $1/\sqrt{k}$. Due to the positive homogeneity of ReLU, these are not two points but two equal-loss hyperbolas in the loss landscape.

5. In Subsection G.5, we study the two-dimensional loss for the softplus activation function and give a proof of Theorem 5.3. Absence of analytical expression for the Gaussian integral terms make the problem challenging; we use several non-trivial steps in the proof. The proof shows that there is no critical point at correlation $-1/\sqrt{k}$; and a non-trivial critical point $(w^*, a^*)$ at correlation $1/\sqrt{k}$ satisfies the following bounds: $a^* \geq k$ and $\|w^*\| \leq 1/\sqrt{k}$. Numerically, we find that these bounds hold for other activation functions studied in this paper (tanh and sigmoid; Fig. 5).

## G.1   Any Non-Trivial Critical Point Satisfies the Lagrangian Condition

We add a reminder here for the definition of the non-trivial critical point $\theta = (w,a)$: it is a critical point of the loss function that satisfies $a \neq 0$ and $\|w\| \neq 0$.

**Proposition G.1.** *Assume that $f^*$ is an orthogonal teacher network of width $k$. If the activation function satisfies Assumption 3.1 (i), any non-trivial critical point $\theta^* = (w^*, a^*)$, i.e. $\nabla L^{1,k}(\theta^*) = 0$, satisfies that $w^*$ is in the span of the teacher's incoming vectors.*

*Proof.* We will prove by contradiction. Let us assume that $w$ is outside of the span of the teacher's incoming vectors. We will show that $(w, a)$ is not a critical point for any $a \neq 0$. Mapping $(w, a)$ to the order parameter space, we get that $(r, u, a)$ where $u = (u_1, ..., u_k) \in \text{int } B$ which implies $u_j \in (-1, 1)$. Since $u \in \text{int } B$ and $r > 0$, we have that $(r, u, a)$ is a critical point of $L_{\text{proj}}^{1,k}$ since the boundaries are not seen near the neighborhood of this point. Therefore the partial derivatives of $L_{\text{proj}}^{1,k}$ are all zero including

$$b_j \frac{d}{du_j} g_\sigma(r, \|v_j\|, u_j) = 0. \tag{104}$$

From Assumption 3.1 (i), we have that $\partial_u g_\sigma(r_1, r_2, u) > 0$ for $u \in (-1, 1)$ which yields a contradiction. Thus, each critical point of the projected loss is on the boundary, i.e. $u \in \partial B$, which implies that the incoming vector is in the span of the teacher's incoming vectors. $\square$

We will next show that any non-trivial critical point satisfies a Lagrangian condition since it is on the boundary of a constrained optimization problem.

**Lemma G.2.** *Let $\theta = (w, a)$ be a non-trivial critical point of $L^{1,k}$. Then the corresponding order parameters $p = (r, u, a)$ satisfy the following Lagrangian condition*

$$b_j \partial_u g_\sigma(r, \|v_j\|, u_j) = \lambda u_j \quad \text{for all } j \in [k]. \tag{105}$$

*Proof.* We will first show that for any differentiable path $(r, \gamma(t), a)$ on the boundary such that $\gamma(t) \in \partial B$ for $t \in (-\epsilon, \epsilon)$ for some $\epsilon > 0$ and $\gamma(0) = u$, the following holds

$$\frac{d}{dt} L_{\text{proj}}^{1,k}(\gamma(t))\big|_{t=0} = \nabla_u L_{\text{proj}}^{1,k}(p) \cdot \gamma'(0) = 0. \tag{106}$$

Let us assume the contrary. We construct the corresponding following path in the weight space

$$\theta(t) = \left( r \left( \sum_{j=1}^{k} u_j(t) v_j + v_\perp \right), a \right). \tag{107}$$

Thanks to the equivalence of the losses along the path, we have that

$$\frac{d}{dt} L^{1,k}(\theta(t))\big|_{t=0} = \frac{d}{dt} L_{\text{proj}}^{1,k}(\gamma(t))\big|_{t=0} = 0, \tag{108}$$

since $\theta(0) = \theta$ is a critical point in the weight space. Therefore, Eq. 106 holds for any differentiable path on the boundary and implies that $\nabla_u L(p)$ is orthogonal to all $\gamma'(0)$. The vector that is orthogonal to all $\gamma'(0)$ is the gradient of the surface, that is $2(u_1, ..., u_k)$. Hence we get $\nabla_u L(p) \parallel u$ which is written explicitly as the Lagrangian condition in Eq. 105. This is equivalent to setting the partial derivatives of the Lagrangian loss with respect to $u_j$ to zero where the Lagrangian loss is given by

$$\mathcal{L}(p, \lambda) = -2a \sum_{j=1}^{k} b_j g_\sigma(r, \|v_j\|, u_j) + \lambda'(\sum_{j=1}^{k} u_j^2 - 1). \tag{109}$$

We set $\lambda = \lambda'/a$ in Eq. 105. $\square$

## G.2 General Activation Functions

Before we present our results, let us take a detour to check the applicability of the convex optimization framework. For a convex and twice-differentiable activation function such as softplus, applying Lemma F.1 twice implies that the interaction $g_\sigma(r_1, r_2, \cdot)$ is a convex function of the correlation $u \in (-1, 1)$ for $r_1, r_2 > 0$. Let us consider a fixed $a < 0$ and $r > 0$ and consider the loss parameterized by $u_j$'s. It is convex since its Hessian is a diagonal matrix with entries

$$\frac{d^2}{du_j^2} L = -2a \frac{d^2}{du_j^2} g_\sigma(r, \|v_j\|, u_j) > 0. \tag{110}$$

Since the constraint on the correlations (Eq. 14) is also convex, we get a convex optimization problem that has a unique global minimum (see Boyd et al. [44], Section 4.2). Swapping a pair of $u_j$ does not change the loss, thus it is permutation symmetric. If any two $u_j$ were distinct from each other at the minimum, then its permutation would also be a minimum which would violate the unicity. We conclude that at the unique minimum point, the correlations are equal to each other. However, for the case $a > 0$, and for other activation functions, the objective is not convex.

We instead use Lagrange multipliers for proving Theorem 5.1.

### G.2.1 Proof of Theorem 5.1

**Theorem G.3.** *Assume that the activation function satisfies Assumption 3.1. At any non-trivial critical point $(w^*, a^*)$ of the loss $L^{1,k}$ for the unit-orthonormal teacher network, the incoming vector satisfies*

$$\frac{w^*}{\|w^*\|} = u \sum_{j=1}^{k} v_j \tag{111}$$

*where $u$ is either $1/\sqrt{k}$ or $-1/\sqrt{k}$.*

*Proof.* From Proposition 4.1 and Lemma G.2, we get that any non-trivial critical point should satisfy the Lagrangian condition in Eq. 105. In particular for unit-orthonormal teacher, setting $\|v_j\| = 1$ and $b_j = 1$, we get the following Lagrangian condition

$$\partial_u g_\sigma(r, 1, u_j) = \lambda u_j \ \forall j \in [k], \quad \sum_{j=1}^{k} u_j^2 = 1. \tag{112}$$

If $u_j = 0$, we get $\partial_u g_\sigma(r, 1, 0) = 0$ which is not possible since $g_\sigma(r, 1, u)$ is increasing due to Assumption 3.1 (i). Hence we have

$$\frac{\partial_u g_\sigma(r, 1, u_j)}{u_j} = \lambda. \tag{113}$$

Let us observe that $\partial_u g_\sigma(r, 1, u)/u$ is decreasing for $u \in (-1, 1) \setminus \{0\}$ if and only if

$$\frac{d}{du}\left(\frac{1}{u}\frac{d}{du}g_\sigma(r, 1, u)\right) = \frac{1}{u}\frac{d^2}{du^2}g_\sigma(r, 1, u) - \frac{1}{u^2}\frac{d}{du}g_\sigma(r, 1, u) < 0, \tag{114}$$

which is equivalent to Assumption 3.1 (ii) for $u \in (-1, 1)\setminus\{0\}$ (we included $u = 0$ in Assumption 3.1 (ii) for a simpler statement which is already implied from Assumption 3.1 (i) at $u = 0$).

Taken together, we conclude that $\partial_u g_\sigma(r, 1, u)/u$ is one-to-one in $u \in (-1, 1) \setminus \{0\}$. We need to consider the remaining case $u_i \in \{-1, 1\}$. For $k \geq 2$, necessarily, we have $u_j = 0$ for $j \neq i$, which is not possible as we have shown. For $k = 1$, $u_i \in \{-1, 1\}$ is the only option that satisfies the boundary condition. For $k \geq 2$, Eq. 113 implies that all correlations are equal. Combining it with the boundary condition, we get $u_1 = ... = u_k = u$ with $ku^2 = 1$, which completes the proof. $\square$

### G.2.2 Two-Dimensional Loss, The Derivative Constraints, Uniqueness

At any non-trivial critical point, we proved in Theorem 5.1 that all correlations are equal and denoted by $u$ that is either $1/\sqrt{k}$ or $-1/\sqrt{k}$. The projected loss at a critical point reduces to

$$L = a^2 g_\sigma(r, r, 1) - 2kag_\sigma(r, 1, u) + C. \tag{115}$$

Moreover, at a critical point, the partial derivatives with respect to the outgoing weight and norm should also be zero which gives the following two constraints

$$\partial_a L = 2ag_\sigma(r, r, 1) - 2kg_\sigma(r, 1, u) = 0,$$
$$\partial_r L = a^2\partial_r g_\sigma(r, r, 1) - 2ka\partial_r g_\sigma(r, 1, u) = 0, \tag{116}$$

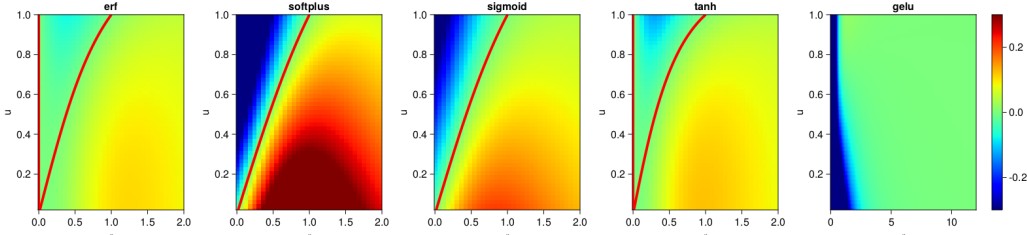

Figure 10: *The graph of $f(r, u) = \frac{d}{dr}\left(\frac{1}{2}\log g_\sigma(r, r, 1) - \log g_\sigma(r, 1, u)\right)$ for activation functions erf, softplus with $\beta = 1$, sigmoid, tanh, and gelu, respectively. Zero crossings of $f$ are shown in red.* For softplus and sigmoid, we observe that $f$ is negative for $r = 0, u \in (0, 1)$, positive for $r = 1, u \in (0, 1)$, and increasing in $r \in [0, 1]$ for any fixed $u$, thus satisfying the sufficient conditions in Eq. 121. However, for tanh and erf, $f$ shows non-monotonic behavior in $r$ when $u$ is close to 1. For the GeLU activation function $\sigma(x) = x\Phi(x)$, which is non-monotonic, we observe that $f$ does not cross zero for any $(r, u)$ pair in the plotted domain. It approaches zero from below when $r \to \infty$ thus showing a very different behavior from the other activation functions.

which can be rearranged into the following (assuming $g_\sigma(r, r, 1) \neq 0$ and $\partial_r g_\sigma(r, r, 1) \neq 0$)

$$\frac{a}{k} = \frac{g_\sigma(r, 1, u)}{g_\sigma(r, r, 1)} = \frac{2\partial_r g_\sigma(r, 1, u)}{\partial_r g_\sigma(r, r, 1)}. \tag{117}$$

The second equality between the two ratios of Gaussian integral terms gives a fixed point equation on the norm $r$. Writing the interactions in Eq. 117 explicitly and rearranging the ratios, we get

$$f(r, u) = \frac{\mathbb{E}[\sigma'(rx)\sigma(rx)x]}{\mathbb{E}[\sigma(rx)^2]} - \frac{\mathbb{E}[\sigma'(rx)\sigma(ry)x]}{\mathbb{E}[\sigma(rx)\sigma(y)]} = 0, \tag{118}$$

where $x$ and $y$ are standard Gaussians with correlation $\mathbb{E}[xy] = u$. Let us define the following helper functions

$$G(r) = \frac{\mathbb{E}[\sigma'(rx)\sigma(rx)x]}{\mathbb{E}[\sigma(rx)^2]} = \frac{1}{2}\frac{d}{dr}\log(\mathbb{E}[\sigma(rx)^2]),$$

$$\tilde{G}(u, r) = \frac{\mathbb{E}[\sigma'(rx)\sigma(y)x]}{\mathbb{E}[\sigma(rx)\sigma(y)]} = \frac{d}{dr}\log(\mathbb{E}[\sigma(rx)\sigma(y)]), \tag{119}$$

which yields

$$f(r, u) = G(r) - \tilde{G}(u, r) = \frac{d}{dr}\log\left(\frac{\mathbb{E}[\sigma(rx)^2]^{\frac{1}{2}}}{\mathbb{E}[\sigma(rx)\sigma(y)]}\right) = 0. \tag{120}$$

Let us consider the case $u > 0$. We want to show that for any given $u \in (0, 1]$ there is a unique $r \in (0, 1)$ such that $f(r, u) = 0$. Under the assumption $\sigma(0) \neq 0$, if the following three conditions are satisfied for all $u \in (0, 1]$,

$$\text{(i)} \quad \frac{\sigma'(0)}{\sigma(0)}\frac{\mathbb{E}[\sigma(y)x]}{\mathbb{E}[\sigma(y)]} > 0,$$

$$\text{(ii)} \quad \frac{\mathbb{E}[\sigma'(x)\sigma(x)x]}{\mathbb{E}[\sigma(x)^2]} > \frac{\mathbb{E}[\sigma'(x)\sigma(y)x]}{\mathbb{E}[\sigma(x)\sigma(y)]},$$

$$\text{(iii)} \quad \frac{d^2}{dr^2}\log\left(\frac{\mathbb{E}[\sigma(rx)^2]^{\frac{1}{2}}}{\mathbb{E}[\sigma(rx)\sigma(y)]}\right) > 0, \tag{121}$$

then we have a unique $r$ solving Eq. 120 as we explain next. Note that the first two conditions are equivalent to $f(0, u) < 0$ and $f(1, u) > 0$, respectively. The tricky part is the third condition which is equivalent to showing that

$$\frac{\mathbb{E}[\sigma(rx)\sigma(y)]}{\mathbb{E}[\sigma(rx)^2]^{\frac{1}{2}}} \tag{122}$$

is log-concave in $r$. We note that marginalization properties of log-concave functions may be helpful here. In this paper, we were not able to prove the sufficient conditions listed above for general activation functions that do not admit an analytic formula of the interaction, even for softplus which we studied in detail (see Subsection G.5). Instead, we present the numerical integration results, which show that for any given $u \in (0, 1]$, there is a unique $r \in (0, 1]$ such that $f = 0$ (see Fig. 10). Once $r$ is shown to be unique, then the matching outgoing weight $a$ follows from Eq. 117.

### G.3 Closed-Form Solution for Erf Activation

**Corollary G.4.** *Assume that the activation function is $\sigma_{erf}$. The optimal solution $(w^*, a^*)$ is given by*

$$\|w^*\| = \sqrt{\frac{1}{2k-1}}, \quad a^* = k, \quad \frac{w^*}{\|w^*\|} = \frac{1}{\sqrt{k}} \sum_{i=1}^{k} v_i,$$

*or, equivalently, by $(-w^*, -a^*)$. The optimal loss is given by*

$$L_{erf}^*(k) = \frac{2}{\pi} \Big( k \arcsin\Big(\frac{1}{2}\Big) - k^2 \arcsin\Big(\frac{1}{2k}\Big) \Big). \tag{123}$$

*Proof.* Since erf is an odd activation function, it suffices to find parameters of the non-trivial critical points satisfying $u \geq 0$. For any such critical point $(w^*, a^*)$, its mirror symmetry $(-w^*, -a^*)$ is an equivalent critical point due to Eq. 41.

For $u = 0$, we have that $g_{\text{erf}}(r, 1, u) = 0$ which implies $g_{\text{erf}}(r, r, 1) = \mathbb{E}[\sigma(rx)^2] = 0$ due to the first derivative constraint in Eq. 116 which holds if and only if $r = 0$. This gives a possible trivial critical point yielding the zero predictor function.

For a given $u = \frac{1}{\sqrt{k}} > 0$, from Fig. 10, we observe that there is a unique $r \in (0, 1]$ satisfying the fixed point equation in Eq. 120. For uniqueness, we rely on numerical integration. We will find one solution to the derivative constraints given below

$$a g_{\text{erf}}(r, r, 1) = k g_{\text{erf}}(r, 1, u), \quad a \partial_r g_{\text{erf}}(r, r, 1) - 2k \partial_r g_{\text{erf}}(r, 1, u) = 0, \tag{124}$$

for a given $k$, and equivalently $u = \frac{1}{\sqrt{k}} > 0$; and due to uniqueness, conclude that it is the only non-trivial critical point up to symmetries.

In particular, we will use the analytic formula for the interaction function [2, 3]

$$g_{\text{erf}}(r_1, r_2, u) = \frac{2}{\pi} \arcsin\Big(\frac{r_1 r_2 u}{\sqrt{r_1^2 + 1}\sqrt{r_2^2 + 1}}\Big). \tag{125}$$

Let us find $r$ where we have

$$g_{\text{erf}}(r, 1, u) = g_{\text{erf}}(r, r, 1)$$

that is satisfied if we have that the arguments of $\arcsin$ match, which happens at

$$\frac{ru}{\sqrt{r^2 + 1}\sqrt{2}} = \frac{r^2}{r^2 + 1} \quad \Rightarrow \quad r = \sqrt{\frac{u^2}{2 - u^2}} = \frac{1}{\sqrt{2k-1}}. \tag{126}$$

Interestingly, at this value of $r$, we also have

$$2\partial_r g_{\text{erf}}(r, 1, u) = \partial_r g_{\text{erf}}(r, r, 1)$$

which can be seen by inserting the guessed values in the following equation

$$2\partial_r \Big(\frac{ru}{\sqrt{r^2 + 1}\sqrt{2}}\Big) \arcsin'\Big(\frac{ru}{\sqrt{r^2 + 1}\sqrt{2}}\Big) = \partial_r \Big(\frac{r^2}{r^2 + 1}\Big) \arcsin'\Big(\frac{r^2}{r^2 + 1}\Big).$$

Setting $a = k$ in Eq. 124 completes the order parameters of the non-trivial critical point. Finally, let us compute the loss at $r = 1/\sqrt{2k-1}, u = 1/\sqrt{k}, a = k$;

$$L_{\text{erf}}^*(k) := a^2 g_{\text{erf}}(r, r, 1) - 2ak g_{\text{erf}}(r, 1, u) + k g_{\text{erf}}(1, 1, 1),$$

$$= -k^2 g_{\text{erf}}(r, r, 1) + k g_{\text{erf}}(1, 1, 1),$$

$$= \frac{2}{\pi} \Big( k \arcsin\Big(\frac{1}{2}\Big) - k^2 \arcsin\Big(\frac{1}{2k}\Big) \Big). \tag{127}$$

$\square$

### G.4 Closed-Form Solution for ReLU Activation

**Corollary G.5.** *Assume that the activation function is $\sigma_{relu}$. Any optimal solution $(w^*, a^*)$ satisfies*

$$\|w^*\|a^* = \frac{k}{h(1)}h\left(\frac{1}{\sqrt{k}}\right), \quad \frac{w^*}{\|w^*\|} = \frac{1}{\sqrt{k}}\sum_{i=1}^{k}v_i, \tag{128}$$

*forming an equal-loss hyperbola. The optimal loss is given by*

$$L_{relu}^*(k) = k^2\left(h(0) - \frac{1}{h(1)}h\left(\frac{1}{\sqrt{k}}\right)^2\right) + k(h(1) - h(0)). \tag{129}$$

We will first show that the interaction of ReLU satisfies

(i) $h'(u) > 0$ for $u \in (-1, 1)$,

(ii) $h''(u)u < h'(u)$ for $u \in (-1, u_0]$,

(iii) $\dfrac{h'(u_0)}{u_0} > \dfrac{h'(u)}{u}$ for $u \in (u_0, 1)$; \hfill (130)

where $u_0 = 1/\sqrt{2}$. Note that property (i) is equivalent to Assumption 3.1 (i), and property (ii) is almost equivalent to Assumption 3.1 (ii) except that it holds in the interval $(-1, u_0]$; property (iii) covers up for the missing piece of the interval in the property (ii).

*ReLU interaction satisfies Properties 130; Proof.* Let us write the first two derivatives of $h$:

$$h'(u) = \frac{\pi - \arccos(u)}{2\pi}, \quad h''(u) = \frac{1}{2\pi\sqrt{1-u^2}}. \tag{131}$$

Property (i) easily comes from noting that the derivative of $h$ is positive for $u \in (-1, 1)$. Property (ii) holds for $u \in (-1, 0]$ since both the first and second derivatives are positive. Let us show that Property (ii) holds for $u \in (0, u_0]$, that is equivalent to

$$\frac{u}{\sqrt{1-u^2}} < \pi - \arccos(u) = \frac{\pi}{2} + \arcsin(u). \tag{132}$$

Let us note that the left-hand side is smaller than $1$ since

$$\frac{u^2}{1-u^2} \leq 1.$$

Note that $\arcsin(u) > 0$ for $u > 0$; and $\pi/2 > 1$. This completes the proof of Property (ii).

For Property (iii), we first show that $h'(u)/u$ is convex in $u \in (0, 1)$. The first two derivatives are

$$\frac{d}{du}\left(\frac{h'(u)}{u}\right) = \frac{h''(u)}{u} - \frac{h'(u)}{u^2}, \quad \frac{d^2}{du^2}\left(\frac{h'(u)}{u}\right) = \frac{h'''(u)}{u} - \frac{2h''(u)}{u^2} + \frac{2h'(u)}{u^3}.$$

Thus, it is equivalent to showing

$$h'''(u)u - 2h''(u) + \frac{2h'(u)}{u} = \frac{u^2}{(1-u^2)^{3/2}} - \frac{2}{(1-u^2)^{1/2}} + \frac{\pi + 2\arcsin(u)}{u} > 0.$$

Using the Taylor series of $\arcsin$ and $u > 0$, we have that $\arcsin(u) > u$. Hence, it suffices to show

$$\frac{1}{(1-u^2)^{1/2}}\left(-3 + \frac{1}{1-u^2} + 2(1-u^2)^{1/2}\right) \geq 0; \tag{133}$$

where we dropped the positive term $\frac{\pi}{u}$ which holds due to the inequality of arithmetic and geometric means

$$\frac{1}{1-u^2} + (1-u^2)^{1/2} + (1-u^2)^{1/2} \geq 3.$$

Let us assume the contrary of Property (iii), that there exists $u \in (u_0, 1)$ such that

$$\frac{h'(u_0)}{u_0} \leq \frac{h'(u)}{u}. \tag{134}$$

Note that $h'(u_0)/u_0 > h'(1)$ because $\pi(1 - u_0) - \arccos(u_0) > 0$ holds at $u_0 = 1/\sqrt{2}$. Since $h'(u)/u$ is left-continuous at $u = 1$, there exists $\epsilon > 0$ such that

$$\frac{h'(u_0)}{u_0} > \frac{h'(1 - \epsilon)}{1 - \epsilon}. \tag{135}$$

Finally, there exists $\alpha \in (0, 1)$ such that $u = \alpha(1 - \epsilon) + (1 - \alpha)u_0$ which gives due to the convexity of $h'(u)/u$ the following

$$\alpha \frac{h'(1 - \epsilon)}{1 - \epsilon} + (1 - \alpha)\frac{h'(u_0)}{u_0} \geq \frac{h'(u)}{u}. \tag{136}$$

This yields a contradiction since the left-hand side is strictly smaller than $h'(u_0)/u_0$ hence the proof of Property (iii) is complete. *ReLU interaction satisfies Properties 130; End of Proof.*

*Proof.* First, we replicate the proof steps of Theorem 5.1 to show that any non-trivial critical point must be on the boundary and attain equal correlations. From Property 130 (i), we get that there is no non-trivial critical point in $\mathrm{int}\, B$. For $k = 1$, this implies that $u_1 = -1$ or $u_1 = 1$.

For general $k$, let us recall that we get the Lagrangian condition for non-trivial critical points

$$rh'(u_j) = \lambda u_j \ \forall j \in [k], \quad \sum_{j=1}^{k} u_j^2 = 1. \tag{137}$$

which is equivalent to Eq. 112 for ReLU activation function. $u_j = 0$ is not possible since we have $h'(0) \neq 0$. Hence, we get

$$\frac{h'(u_j)}{u_j} = \frac{\lambda}{r}, \quad \forall j \in [k]. \tag{138}$$

Property 130 (ii) implies that $f(u) = h'(u)/u$ is decreasing for $u \in (-1, u_0)\backslash\{0\}$. Moreover, $f$ is negative for $u < 0$ and positive for $u > 0$.

1. If $\lambda/r < 0$, we get that all $u_j$ are equal and negative, hence they are equal to $-1/\sqrt{k}$ due to the boundary condition.

2. If $\lambda/r = 0$, we get $u_j = -1$ for all $j$ which implies that $k = 1$ which is already covered above.

3. If $\lambda/r > 0$, Property 130 (iii) gives that $f(u_0) > f(u)$ for $u \in (u_0, 1)$. Since $f$ is decreasing we have also $f(u) > f(u_0)$ for $u \in (0, u_0)$; hence $f(u_j)$ are equal only when all $u_j < u_0$ or $u_j > u_0$; however, the latter case is not possible for $k \geq 2$ since it breaks the ball constraint, i.e. $u_1^2 + u_2^2 > 1$.

Hence, we get that $u_j \in (0, u_0]$ and are equal since $f$ is decreasing in this interval. This completes the proof of replica of Theorem 5.1 for the ReLU activation function.

For the ReLU activation function, there is at least one non-differentiable critical point at $a = 0$ or $r = 0$. The careful analysis of this critical point is beyond the scope of this work. For any such 'trivial' point, the error of zero-function is equivalent to

$$\mathbb{E}[(\sum_{j=1}^{k} \sigma(v_j \cdot x))^2] = kh(1) + k(k - 1)h(0). \tag{139}$$

We will next show that $(-1/\sqrt{k})_{j=1}^{k}$ and $(1/\sqrt{k})_{j=1}^{k}$ are the global minimum and the global maximum of the following loss function

$$\sum_{j=1}^{k} h(u_j), \quad \text{subject to} \sum_{j=1}^{k} u_j^2 \leq 1. \tag{140}$$

Due to the Lagrange condition, there is no other critical point, hence these are the only two critical points of the constrained objective in Eq. 140. The objective then reduces to $kh(u)$ which is minimized at $u = -1/\sqrt{k}$ and maximized at $u = 1/\sqrt{k}$.

Next, we will give the closed-form solution of the remaining order parameters. Plugging in the correlation in the loss and using the factorization of the interaction in Eq. 14, we get

$$L = a^2 r^2 \cdot h(1) - 2kar \cdot h(u) + C.$$

Let us set $\tilde{a} = ar$. The loss is a second-order polynomial in $\tilde{a}$

$$L = h(1)\left(\tilde{a}^2 - 2\tilde{a}k\frac{h(u)}{h(1)} + k + k(k-1)\frac{h(0)}{h(1)}\right)$$

where we made the constant explicit. Since the coefficient of the leading term is positive, there is a minimizer and it is the only critical point. Taking the derivative, the minimum is attained at

$$\tilde{a}_* = k\frac{h(u)}{h(1)} \tag{141}$$

Finally, plugging in $\tilde{a}_*$, we get

$$L(u) = -k^2\frac{h(u)^2}{h(1)} + kh(1) + k(k-1)h(0). \tag{142}$$

For $u = 1/\sqrt{k}$ and $u = -1/\sqrt{k}$, $h(u)$ is non-zero; hence $l(u)$ is smaller than the loss of the zero function (trivial critical points). The smallest loss is attained at $u = 1/\sqrt{k}$ which is, therefore, the optimal solution. We conclude that the critical point at $u = -1/\sqrt{k}$ is a saddle point since it is a maximum in $u$ and a minimum in $\tilde{a}$. $\qquad\square$

### G.5 Bounds on Incoming Vector Norm and Outgoing Weight for Softplus

Unlike ReLU and erf, the interaction function does not have a known analytic expression for softplus, hence the proof involves some techniques to compare ratios of Gaussian integral terms.

*FKG Inequality.* We will use a special case of the FKG inequality repeatedly, that is,

$$\mathbb{E}[f(x)g(x)] > \mathbb{E}[f(x)]\mathbb{E}[g(x)] \tag{143}$$

if both $f, g$ are increasing (or decreasing) implying that $f$ and $g$ are positively correlated. The inequality changes direction if $f$ is increasing and $g$ is decreasing (or vice versa) implying that $f$ and $g$ are negatively correlated.

We will rely on some specific properties of the softplus family that are developed in Section G.5.4. Unfortunately, some of these properties do not apply to other activation functions. As a first example of managing interactions that do not have an analytic formula, the proof may inspire generalizations to other activation functions. Below we present the proof sketch for Theorem 5.3. In the following Subsections G.5.1, G.5.2, G.5.3, and G.5.4, the components of the proof are presented in detail.

*Proof Sketch.* We want to characterize the zero(s) of $f$ introduced in Section G.2.2 that is

$$f(r, u) = G(r) - \tilde{G}(u, r) = \frac{\mathbb{E}[\sigma'(rx)\sigma(rx)x]}{\mathbb{E}[\sigma(rx)^2]} - \frac{\mathbb{E}[\sigma'(rx)\sigma(y)x]}{\mathbb{E}[\sigma(rx)\sigma(y)]}. \tag{144}$$

For $r \in [0, 1]$, there is a unique correlation $u \in [0, 1]$ such that $f(r, u) = 0$. Denoting this correlation by $h(r)$, we have a map $h : [0, 1] \to [0, 1]$ with boundary conditions $h(0) = 0$ and $h(1) = 1$. For $r > 1$, there is no solution of $f$. As a consequence, no $r \geq 0$ solves $f(r, u) = 0$ for negative $u$, hence there is no non-trivial critical point at $u = -1/\sqrt{k}$.

In Section G.5.2, we prove the inequality $h(r) \geq r$, which gives us the upper bound on the norm since we have that the correlation at a non-trivial critical point is $h(r) = 1/\sqrt{k}$. Using this inequality and Stein's Lemma, we give a lower bound on the outgoing weight, that is $a \geq k$, in Section G.5.3. In summary, any non-trivial critical point of the loss has equal correlations that are $u = 1/\sqrt{k}$, the norm satisfies $r \leq u$, and the lower bound on the outgoing weight follows. *End of Proof Sketch.*

### G.5.1 Constraining the Zeros of $f$

In this subsection, we will describe all zero-crossings of $f : [0, \infty) \times [-1, 1] \to \mathbb{R}$. We need to check four cases *(i)* $r = 0$, *(ii)* $r = 1$, *(iii)* $r > 1$, and *(iv)* $r \in (0, 1)$.

*(i)* $r = 0$: Note that $G(0) = \tilde{G}(0, 0) = 0$. Since $\tilde{G}$ is increasing in correlation for $u \in [0, 1]$ and $\tilde{G}(u, 0) < \tilde{G}(0, 0)$ for $u < 0$ (Lemma G.6), the only solution is $u = 0$.

*(ii)* $r = 1$: Note that $G(1) = \tilde{G}(1, 1)$ since $y = x$ due to correlation one in Eq. 119. Since $\tilde{G}$ is increasing in correlation for $u \in [0, 1]$ and $\tilde{G}(u, 1) < \tilde{G}(0, 1)$ for $u < 0$ (Lemma G.6), the only solution is $u = 1$.

*(iii)* $r > 1$: We will show that there is no zero in this case. Let us first show that $G(r) > \tilde{G}(1, r)$ for $r > 1$, which is equivalent to

$$\mathbb{E}[\sigma'(rx)\sigma(rx)x]\mathbb{E}[\sigma(rx)\sigma(x)] > \mathbb{E}[\sigma'(rx)\sigma(x)x]\mathbb{E}[\sigma(rx)^2]. \tag{145}$$

Changing the measure of $x$ from the standard Gaussian $p(x)$ to $\tilde{p}(x) = p(x)\sigma(rx)^2/\mathbb{E}[\sigma(rx)^2]$, we get the following equivalent inequality

$$\mathbb{E}_{x \sim \tilde{p}}\left[\frac{\sigma'(rx)x}{\sigma(rx)}\right] \mathbb{E}_{x \sim \tilde{p}}\left[\frac{\sigma(x)}{\sigma(rx)}\right] > \mathbb{E}_{x \sim \tilde{p}}\left[\frac{\sigma'(rx)x}{\sigma(rx)} \frac{\sigma(x)}{\sigma(rx)}\right]. \tag{146}$$

From the property (iv) of Lemma G.8, we have that $\sigma'(rx)x/\sigma(rx)$ is increasing after a substitution $x \leftarrow rx$. We need to show $\sigma(x)/\sigma(rx)$ is decreasing in $x$ for $r > 1$. We take the derivative

$$\frac{d}{dx}\frac{\sigma(x)}{\sigma(rx)} = \frac{\sigma'(x)\sigma(rx) - \sigma(x)\sigma'(rx)r}{\sigma(rx)^2}. \tag{147}$$

Since $\sigma'(x)x/\sigma(x)$ is increasing $\forall x \in \mathbb{R}$, we have

$$\frac{\sigma'(x)x}{\sigma(x)} < \frac{\sigma'(rx)rx}{\sigma(rx)} \text{ for } x > 0, \text{ and } \frac{\sigma'(x)x}{\sigma(x)} > \frac{\sigma'(rx)rx}{\sigma(rx)} \text{ for } x < 0$$

which yields $\sigma'(x)\sigma(rx) < \sigma(x)\sigma'(rx)r$, hence we conclude that $\sigma(x)/\sigma(rx)$ is decreasing. Thanks to the FKG inequality, $\sigma'(rx)x/\sigma(rx)$ and $\sigma(x)/\sigma(rx)$ are negatively correlated which completes the argument. Since from Lemma G.6, $\tilde{G}$ is increasing in correlation and $\tilde{G}(0, r) > \tilde{G}(u, r)$ for $u < 0$, we have $\tilde{G}(1, r) > \tilde{G}(u, r)$ for all $u \in [-1, 1)$, therefore there is no solution of $f$.

*(iv)* $r \in (0, 1)$: We want to show that $\forall r \in (0, 1)$, there is a unique $u \in (0, 1)$ such that $f(r, u) = 0$. It suffices to show

$$\tilde{G}(0, r) < G(r) < \tilde{G}(1, r),$$

since $\tilde{G}$ is continuous and increasing in correlation for $u \in [0, 1]$ (Lemma G.6), it then crosses $G(r)$ at a unique $u \in (0, 1)$.

*First inequality;* $\tilde{G}(0, r) < G(r)$. In this case, $x$ and $y$ are Gaussians with zero correlation, hence independent. We can expand $\tilde{G}(0, r)$ by factorizing the integrals

$$\tilde{G}(0, r) = \frac{\mathbb{E}\left[\sigma'(rx)x\right]\mathbb{E}\left[\sigma(y)\right]}{\mathbb{E}\left[\sigma(rx)\right]\mathbb{E}\left[\sigma(y)\right]} = \frac{\mathbb{E}\left[\sigma'(rx)x\right]}{\mathbb{E}\left[\sigma(rx)\right]}.$$

We want to show

$$\mathbb{E}\left[\sigma'(rx)x\right]\mathbb{E}\left[\sigma(rx)^2\right] < \mathbb{E}\left[\sigma'(rx)\sigma(rx)x\right]\mathbb{E}\left[\sigma(rx)\right] \tag{148}$$

which is equivalent to the following inequality after changing the measure from standard Gaussian $p(x)$ to $\tilde{p}(x) = p(x)\sigma(rx)/\mathbb{E}[\sigma(rx)]$

$$\mathbb{E}_{x \sim \tilde{p}}\left[\frac{\sigma'(rx)x}{\sigma(rx)}\right] \mathbb{E}_{x \sim \tilde{p}}[\sigma(rx)] < \mathbb{E}_{x \sim \tilde{p}}\left[\frac{\sigma'(rx)x}{\sigma(rx)}\sigma(rx)\right]. \tag{149}$$

This follows from the FKG inequality since we have that both $\sigma'(x)x/\sigma(x)$ and $\sigma(x)$ are increasing from the properties (iv) and (i) of softplus (Lemma G.8).

*Second inequality;* $G(r) < \tilde{G}(1, r)$. This is equivalent to the Ineq. 145, but the direction is reversed since in this case $r < 1$. We showed that $\sigma(x)/\sigma(r'x)$ is decreasing in $x$ for all $r' > 1$, therefore its reciprocal $\sigma(r'x)/\sigma(x)$ is increasing in $x$. Substituting $x \leftarrow rx$ where $r = 1/r' < 1$, we get that $\sigma(x)/\sigma(rx)$ is increasing in $x$ for $r < 1$. This yields a positive correlation between $\sigma'(rx)x/\sigma(rx)$ and $\sigma(x)/\sigma(rx)$ from the FKG inequality and completes the argument.

Overall, we showed that there are no zeros of $f$ for $r > 1$. For $r \in [0, 1]$, there is a unique correlation $u$, that we will denote by $h(r)$, such that $f(r, h(r)) = 0$. Furthermore, $h : [0, 1] \rightarrow [0, 1]$ satisfies the following

i. $h(0) = 0$ and $h(1) = 1$,

ii. for $r \in (0, 1)$, we have $h(r) \in (0, 1)$.

### G.5.2 Bound on the Norm

In this subsection, we will show that $h(r) \geq r$ for all $r \in (0, 1)$. Let us assume the contrary, which implies

$$\tilde{G}(h(r), r) < \tilde{G}(r, r)$$

due to Lemma G.6. It suffices to show that for all $r \in (0, 1)$, we have

$$\tilde{G}(r, r) \leq G(r), \tag{150}$$

which yields a contradiction since $G(r) = \tilde{G}(h(r), r)$. Showing this is equivalent to

$$\mathbb{E}\left[\sigma'(rx)\sigma(rx + r'z)x\right]\mathbb{E}\left[\sigma(rx)^2\right] \leq \mathbb{E}\left[\sigma'(rx)\sigma(rx)x\right]\mathbb{E}\left[\sigma(rx)\sigma(rx + r'z)\right] \tag{151}$$

where $r' = \sqrt{1 - r^2}$. After a change of measure from standard Gaussian $p(x)$ to

$$\tilde{p}(x) = p(x)\frac{\mathbb{E}[\sigma(rx + r'z)|x]\sigma(rx)}{\mathbb{E}\left[\sigma(rx + r'z)\sigma(rx)\right]},$$

this is equivalent to the following inequality

$$\mathbb{E}_{x \sim \tilde{p}}\left[\frac{\sigma'(rx)x}{\sigma(rx)}\right]\mathbb{E}_{x \sim \tilde{p}}\left[\frac{\sigma(rx)}{\mathbb{E}[\sigma(rx + r'z)|x]}\right] \leq \mathbb{E}_{x \sim \tilde{p}}\left[\frac{\sigma'(rx)x}{\sigma(rx)}\frac{\sigma(rx)}{\mathbb{E}[\sigma(rx + r'z)|x]}\right]. \tag{152}$$

What remains to show is that

$$\frac{\mathbb{E}[\sigma(rx + r'z)|x]}{\sigma(rx)}$$

is non-increasing in $x$ since then we can conclude by the FKG inequality. Since $r > 0$ we can drop it up to a change in the standard deviation of $x$. We want to show that its derivative is non-positive:

$$\sigma(x)\mathbb{E}[\sigma'(x + r'z)|x] \leq \sigma'(x)\mathbb{E}[\sigma(x + r'z)|x] \Leftrightarrow \frac{\sigma(x)}{\sigma'(x)} \leq \frac{\mathbb{E}[\sigma(x + r'z)|x]}{\mathbb{E}[\sigma'(x + r'z)|x]}. \tag{153}$$

From the property (iii) of softplus (Lemma G.8), we have that $R(x) = \sigma(x)/\sigma'(x)$ is convex. Applying Jensen, we get

$$\frac{\sigma(x)}{\sigma'(x)} \leq \mathbb{E}\left[\frac{\sigma(x + r'z)}{\sigma'(x + r'z)}\Big|x\right].$$

What remains to show is that

$$\mathbb{E}\left[\frac{\sigma(x + r'z)}{\sigma'(x + r'z)}\Big|x\right]\mathbb{E}\left[\sigma'(x + r'z)|x\right] \leq \mathbb{E}\left[\sigma(x + r'z)|x\right]. \tag{154}$$

Note that $\mathbb{E}[\sigma'(x + r'z)|x]$ is increasing in $x$ since $\sigma'$ is increasing. Moreover, the function

$$\mathbb{E}\left[\frac{\sigma(x + r'z)}{\sigma'(x + r'z)}\Big|x\right]$$

is increasing in $x$ since its integrand $R$ is increasing from the property (ii) of softplus (Lemma G.8). Then we conclude by the FKG inequality that Eq. 154 holds. Therefore, for a solution $(r, u)$ of the fixed point Eq. 117, we have $r \leq u = \frac{1}{k}$.

### G.5.3 Bounding the Outgoing Weight

To get a bound on $a$, let us analyze the ratio of interactions in Eq. 117

$$\frac{g_\sigma(r, 1, u)}{g_\sigma(r, r, 1)} = \frac{a}{k}. \tag{155}$$

Using the convexity of softplus (property (i) of Lemma G.8), we get

$$\frac{\mathbb{E}[\sigma(rx)\sigma(ux + u'z)]}{\mathbb{E}[\sigma(rx)^2]} \geq \frac{\mathbb{E}[\sigma(rx)\sigma(rx)] + \mathbb{E}[\sigma(rx)((u-r)x + u'z)\sigma'(rx)]}{\mathbb{E}[\sigma(rx)^2]}$$

$$= 1 + (u - r)\frac{\mathbb{E}[\sigma'(rx)\sigma(rx)x]}{\mathbb{E}[\sigma(rx)^2]}. \tag{156}$$

We can transform the numerator using Stein's lemma with $v(x) = \sigma(rx)\sigma'(rx)$

$$\mathbb{E}[\sigma(rx)\sigma'(rx)x] = r\mathbb{E}[\sigma'(rx)^2 + \sigma(rx)\sigma''(rx)] \tag{157}$$

which is positive since softplus is positive, increasing, and convex. Combining it with $u \geq r$, we get that the ratio is bounded below by 1 which yields $a \geq k$.

### G.5.4 Helper Lemmas

In this subsection, we provide helper lemmas used in the proof of Theorem 5.3. We present Lemma G.6 which shows that $\tilde{G}$ is increasing in correlation and Lemma G.7 used in the proof of the former. Finally, we present several properties of the softplus family in Lemma G.8 that are used throughout the proof.

**Lemma G.6.** *The following function is increasing in* $u \in [0, 1]$

$$\tilde{G}(u, r) = \frac{\mathbb{E}[\sigma'(rx)\sigma(y)x]}{\mathbb{E}[\sigma(rx)\sigma(y)]} \tag{158}$$

*for any* $r \geq 0$, *where* $x$ *and* $y$ *are standard Gaussians with correlation* $\mathbb{E}[xy] = u$. *Moreover,* $\tilde{G}(u, r) < \tilde{G}(0, r)$ *for* $u < 0$.

*Proof.* Let us assume $0 \leq u_1 < u_2 \leq 1$. For the first part of the statement, we want to show

$$\frac{\mathbb{E}[\sigma'(rx)\sigma(y_1)x]}{\mathbb{E}[\sigma(rx)\sigma(y_1)]} < \frac{\mathbb{E}[\sigma'(rx)\sigma(y_2)x]}{\mathbb{E}[\sigma(rx)\sigma(y_2)]} \tag{159}$$

where $\mathbb{E}[xy_1] = u_1$ and $\mathbb{E}[xy_2] = u_2$. Changing the measure from the standard Gaussian $p(x)$ to

$$\tilde{p}(x) = p(x)\frac{\sigma(rx)\mathbb{E}[\sigma(y_2)|x]}{\mathbb{E}[\sigma(rx)\sigma(y_2)]},$$

we get the following equivalent inequality

$$\mathbb{E}\left[\frac{\sigma'(rx)x}{\sigma(rx)}\frac{\mathbb{E}[\sigma(y_1)|x]}{\mathbb{E}[\sigma(y_2)|x]}\right] < \mathbb{E}\left[\frac{\sigma'(rx)x}{\sigma(rx)}\right]\mathbb{E}\left[\frac{\mathbb{E}[\sigma(y_1)|x]}{\mathbb{E}[\sigma(y_2)|x]}\right]. \tag{160}$$

Thanks to the property (iv) of softplus (Lemma G.8), we have that $\sigma'(rx)x/\sigma(rx)$ is increasing in $x$ after a substitution $x \leftarrow rx$ for $r > 0$. For $r = 0$, the function reduces to $\gamma x$ with some $\gamma > 0$, hence increasing. We will next show that (all integrations are w.r.t $z$ hereafter, hence we drop the conditioning on $x$)

$$\frac{\mathbb{E}[\sigma(y_1)]}{\mathbb{E}[\sigma(y_2)]} \tag{161}$$

is decreasing in $x$. Computing the derivative w.r.t $x$, we want to show that it is negative

$$\frac{\mathbb{E}[\sigma'(y_1)]u_1}{\mathbb{E}[\sigma(y_2)]} - \frac{\mathbb{E}[\sigma(y_1)]\mathbb{E}[\sigma'(y_2)]u_2}{\mathbb{E}[\sigma(y_2)]^2} < 0 \quad \Leftrightarrow \quad \frac{\mathbb{E}[\sigma'(y_1)]u_1}{\mathbb{E}[\sigma(y_1)]} < \frac{\mathbb{E}[\sigma'(y_2)]u_2}{\mathbb{E}[\sigma(y_2)]}. \tag{162}$$

Note that this is equivalent to showing

$$\frac{d}{du} \frac{\mathbb{E}[\sigma'(y)]u}{\mathbb{E}[\sigma(y)]} = \frac{d^2}{dudx} \log(\mathbb{E}[\sigma(y)]) > 0$$

for all $u \in [0,1)$ and $x \in \mathbb{R}$. Changing the order of derivatives, it is sufficient to show

$$\frac{d}{dx} \left( \frac{\mathbb{E}[\sigma'(y)]x}{\mathbb{E}[\sigma(y)]} - \left(\frac{u}{1-u^2}\right) \frac{\mathbb{E}[u'z\sigma'(y)]}{\mathbb{E}[\sigma(y)]} \right) > 0. \tag{163}$$

The first function

$$s_1(x) = \frac{\mathbb{E}[\sigma'(y)]x}{\mathbb{E}[\sigma(y)]} \tag{164}$$

is shown to be increasing in $x$ in Lemma G.7 where we need to substitute $x \to xu_1$ for $u_1 > 0$, and for $u_1 = 0$, we have $s_1(x) = \gamma x$ for some $\gamma > 0$ hence it is increasing. The remaining part is to show that the second function

$$s_2(x) = \frac{\mathbb{E}[u'z\sigma'(y)]}{\mathbb{E}[\sigma(y)]} \tag{165}$$

is decreasing. We will consider $z \leftarrow u'z$ and $x \leftarrow ux$ in what follows. We have

$$\frac{d}{dx} \frac{\mathbb{E}[z\sigma'(x+z)]}{\mathbb{E}[\sigma(x+z)]} < 0 \quad \Leftrightarrow \quad \frac{d}{dx} \frac{\mathbb{E}[\sigma(x+z)]}{\mathbb{E}[\sigma''(x+z)]} > 0$$

due to first applying Stein's Lemma to the numerator and then inverting the ratio. Using the chain rule, it is sufficient to show that

$$f_1(x) = \frac{\mathbb{E}[\sigma(x+z)]}{\mathbb{E}[\sigma'(x+z)]}, \quad \text{and} \quad f_2(x) = \frac{\mathbb{E}[\sigma'(x+z)]}{\mathbb{E}[\sigma''(x+z)]} \tag{166}$$

are increasing, since both functions are positive.

Interestingly, $f_1$ is increasing in $x$ if $\sigma$ is a log-concave function. Because its derivative is positive

$$\frac{d}{dx} f_1(x) = 1 - \frac{\mathbb{E}[\sigma(x+z)]\mathbb{E}[\sigma''(x+z)]}{\mathbb{E}[\sigma'(x+z)]^2} > 0$$

if $\mathbb{E}[\sigma(x+z)]$ is log-concave. This is the case since a centered Gaussian distribution $p(z)$ is log-concave, therefore $\sigma(x+z)p(z)$ is jointly log-concave, and marginalization preserves log-concavity.

Similarly, $f_2$ is increasing since $\sigma'$ is also log-concave due to property (v) of softplus (Lemma G.8). Hence we showed that

$$r(u) = \frac{\mathbb{E}[\sigma'(y)]x}{\mathbb{E}[\sigma(y)]} \tag{167}$$

is increasing for $u \in [0,1)$. The derivative of $r$ explodes at 1, however, we can conclude by contradiction that $r(1) > r(u)$ for $u < 1$: if $r(u) \geq r(1)$ for some $0 \leq u < 1$, then there exists $u_0 \in (u,1)$ where the function is decreasing. Therefore, $r$ is increasing for $u \in [0,1]$. We can conclude the first part of the proof by the FKG inequality $\sigma'(rx)x/\sigma(rx)$ and $\mathbb{E}[\sigma(y_1)]/\mathbb{E}[\sigma(y_2)]$ are negatively correlated.

For the second part of the statement, we need to show

$$\mathbb{E}[\sigma'(rx)\sigma(ux+u'z)x]\mathbb{E}[\sigma(rx)] < \mathbb{E}[\sigma'(rx)x]\mathbb{E}[\sigma(rx)\sigma(ux+u'z)] \tag{168}$$

for $u < 0$. Changing the measure from standard Gaussian $p(x)$ to

$$\tilde{p}(x) = p(x)\frac{\sigma(rx)}{\mathbb{E}[\sigma(rx)]}, \tag{169}$$

the above inequality is equivalent to

$$\mathbb{E}_{x\sim\tilde{p}}\left[\frac{\sigma'(rx)x}{\sigma(rx)}\sigma(ux+u'z)\right] < \mathbb{E}_{x\sim\tilde{p}}\left[\frac{\sigma'(rx)x}{\sigma(rx)}\right]\mathbb{E}_{x\sim\tilde{p}}[\sigma(ux+u'z)]. \tag{170}$$

This holds since $\sigma'(rx)x/\sigma(rx)$ is increasing in $x$, however, $\sigma(ux+u'z)$ is decreasing in $x$ since $u$ is negative which implies a negative correlation due to the FKG inequality. $\square$

**Lemma G.7.** *The following function*

$$\frac{\mathbb{E}[\sigma'(x+z)|x]x}{\mathbb{E}[\sigma(x+z)|x]}$$

*is increasing in $x$ where the integrations are w.r.t a centered Gaussian $z$.*

*Proof.* Since all integrals are w.r.t $z$, we drop the conditioning with respect to $x$ in the proof. Taking the derivative w.r.t $x$, and arranging the terms, it suffices to show

$$\left(\frac{\mathbb{E}[\sigma''(x+z)]x}{\mathbb{E}[\sigma'(x+z)]}+1\right)\mathbb{E}[\sigma(x+z)] > \mathbb{E}[\sigma'(x+z)]x \tag{171}$$

which is equivalent to the following due to the property $\sigma''(z) = \beta\sigma'(z)(1-\sigma'(z))$

$$\left(\beta x\left(1-\frac{\mathbb{E}[\sigma'(x+z)^2]}{\mathbb{E}[\sigma'(x+z)]}\right)+1\right)\mathbb{E}[\sigma(x+z)] > \mathbb{E}[\sigma'(x+z)]x. \tag{172}$$

In the case $x \geq 0$, the LHS is bigger than $\mathbb{E}[\sigma(x+z)]$ since $\sigma'(\cdot)$ is upper bounded by 1. Moreover, since $\sigma(x) > x$ and from the convexity of softplus, we get $\mathbb{E}[\sigma(x+z)] > x$. This yields the above inequality by again noting that $\mathbb{E}[\sigma'(x+z)]$ is upper bounded by 1.

In the case $x < 0$, we need another strategy. We have thanks to Cauchy-Schwartz

$$\frac{\mathbb{E}[\sigma'(x+z)^2]}{\mathbb{E}[\sigma'(x+z)]} \geq \mathbb{E}[\sigma'(x+z)], \tag{173}$$

thus it suffices to show

$$(\beta x - \beta x \mathbb{E}[\sigma'(x+z)]+1)\mathbb{E}[\sigma(x+z)] > \mathbb{E}[\sigma'(x+z)]x. \tag{174}$$

We will now show the following

$$\mathbb{E}[\sigma'(x+z)] \geq \sigma'(x) \tag{175}$$

for which it suffices to show that $v(z) := \sigma'(x+z) + \sigma'(x-z) \geq 2\sigma'(x)$ for all $z$ since the centered Gaussian measure $p(z)$ is even and the integration can be done over the integrand $v(z)$. We have

$$v'(z) = \sigma''(x+z) - \sigma''(x-z) \tag{176}$$

that is zero iff either $x+z = x-z$ or $x+z = -x+z$ where the latter is not possible since $x < 0$. Hence we get that a critical point of $v(z)$ at $z = 0$ which is a minimizer since $v''(0) = 2\sigma'''(x) > 0$ for $x < 0$. Hence $v(z) \geq v(0) = 2\sigma'(x)$ for all $z$ which completes the argument.

Finally, it remains to show

$$\left(\frac{\beta x}{e^{\beta x}+1}+1\right)\mathbb{E}[\sigma(x+z)] > \mathbb{E}[\sigma'(x+z)]x. \tag{177}$$

From the proof of Lemma G.8, we have that $\beta\sigma(x) > \sigma'(x)$, which in combination with the following trivial observation for all $x < 0$ (note that $+1$ is not needed for the following to hold)

$$\frac{\beta x}{e^{\beta x}+1}+1 > \beta x \tag{178}$$

shows that Eq. 177 holds, hence the proof is complete. $\square$

**Lemma G.8.** *The softplus family has the following properties*

  *i. $\sigma(x)$ is increasing and convex,*

  *ii. $\sigma(x)$ is log-concave (equivalently, $\sigma(x)/\sigma'(x)$ is increasing),*

  *iii. $\sigma(x)/\sigma'(x)$ is convex,*

  *iv. $\sigma'(x)x/\sigma(x)$ is increasing,*

  *v. $\sigma'(x)$ is log-concave.*

*Proof.* For the property (i), see the formulas of $\sigma'$ and $\sigma''$ in the proof of Lemma F.2. We next prove each one of the properties one after the other. Let us start with property (ii). First note that $\sigma(x)$ is log-concave if and only if $\sigma(x)/\sigma'(x)$ is increasing since

$$\frac{d}{dx}\frac{\sigma(x)}{\sigma'(x)} = 1 - \frac{\sigma(x)\sigma''(x)}{\sigma'(x)^2} > 0 \Leftrightarrow \sigma'(x)^2 > \sigma(x)\sigma''(x) \tag{179}$$

where the second inequality is a characterization of log-concavity. We will prove that $R(x) := \sigma(x)/\sigma'(x)$ is increasing.

Let us write out the ratio explicitly

$$R(x) = \frac{1}{\beta}\left(\log(e^{\beta x} + 1) + \frac{\log(e^{\beta x} + 1)}{e^{\beta x}}\right). \tag{180}$$

The first derivative of $R$ is given by

$$R'(x) = \sigma'(x) + \frac{\sigma'(x) - \beta\sigma(x)}{e^{\beta x}} = \frac{e^{\beta x} - \beta\sigma(x)}{e^{\beta x}}. \tag{181}$$

Since log is concave, expanding it around 1 we get $\log(y+1) < y$ for all $y > 0$. Substituting $y = e^{\beta x}$, we get that the numerator of $R'$ is positive, thus $R$ is increasing. This completes the proof of property (ii). Computing the second derivative of $R$, we get

$$R''(x) = \frac{\sigma''(x)(e^{\beta x} + 1) - 2\beta\sigma'(x) + \beta^2\sigma(x)}{e^{\beta x}} = \beta\left(\frac{-\sigma'(x) + \beta\sigma(x)}{e^{\beta x}}\right). \tag{182}$$

What remains to show is that $\beta\sigma(x) > \sigma'(x)$. Using the fundamental theorem of calculus, we get

$$\frac{\log(y+1)}{y} = \frac{1}{y}\int_0^y \frac{1}{t+1}dt > \frac{1}{y+1} \tag{183}$$

since $1/(y+1)$ is a lower bound of the integrand which completes the proof of the property (iii). Let us prove the property (iv) by taking the derivative of the function of interest

$$\frac{d}{dx}\frac{\sigma'(x)x}{\sigma(x)} = \frac{(\sigma''(x)x + \sigma'(x))\sigma(x) - \sigma'(x)^2 x}{\sigma(x)^2} \tag{184}$$

Using $\sigma''(x) = \beta\sigma'(x)(1 - \sigma'(x))$ and dropping the positive term $\sigma'(x)$, the numerator of the derivative is

$$((1 - \sigma'(x))\beta x + 1)\sigma(x) - \sigma'(x)x = \left(\frac{\beta x}{e^{\beta x} + 1} + 1\right)\sigma(x) - \frac{e^{\beta x}}{e^{\beta x} + 1}x \tag{185}$$

$$= \frac{e^{\beta x}}{e^{\beta x} + 1}\left(\frac{1}{e^{\beta x}}(\beta x + e^{\beta x} + 1)\sigma(x) - x\right) \tag{186}$$

For the case $x \geq 0$, we have $\sigma(x) > x$ and $(\beta x + 1)/e^{\beta x} > 0$, hence the derivative is positive. For the case $x < 0$, we want to show

$$\left(e^{\beta x} + \beta x + 1\right)\frac{\log(e^{\beta x} + 1)}{e^{\beta x}} > \beta x. \tag{187}$$

If $e^{\beta x} + \beta x + 1 > 0$, it is done since the LHS is positive. If $e^{\beta x} + \beta x + 1 \leq 0$, we have

$$\left(e^{\beta x} + \beta x + 1\right)\frac{\log(e^{\beta x} + 1)}{e^{\beta x}} \geq \left(e^{\beta x} + \beta x + 1\right)\sup\frac{\log(e^{\beta x} + 1)}{e^{\beta x}} \tag{188}$$

since $\log(e^{\beta x} + 1)/e^{\beta x}$ is positive. We will next show that $\log(e^{\beta x} + 1)/e^{\beta x}$ is a decreasing function therefore its supremum is achieved at $x \to -\infty$. From the integral expression in Eq. 183, we deduce that $\log(y+1)/y$ is a decreasing function since adding smaller terms in the average decreases it. Thus the following limit gives us the supremum using L'Hôpital's rule

$$\lim_{y \to 0}\frac{\log(y+1)}{y} = \lim_{y \to 0}\frac{1}{y+1} = 1. \tag{189}$$

Combining it with the Eq. 188 after the substitution $y = e^{\beta x}$, we get the desired Ineq. 187 which implies that the derivative is positive in this case too. This completes the proof of property (iv).

For the property (v), we first give a formula for the third derivative of softplus

$$\sigma'''(x) = \beta\sigma''(x)(1 - 2\sigma'(x)). \tag{190}$$

$\sigma'$ is log-concave if and only if we have

$$\sigma'''(x)\sigma(x) < \sigma''(x)\sigma'(x) \Leftrightarrow$$
$$\beta\sigma''(x)(1 - 2\sigma'(x))\sigma(x) < \sigma''(x)\sigma'(x) \tag{191}$$

which is equivalent to

$$(1 - e^{\beta x})\log(e^{\beta x} + 1) < e^{\beta x}. \tag{192}$$

This is equivalent to $(1 - y)\log(y + 1) < \log(y + 1) < y$ where $y = e^{\beta x} > 0$; the second inequality holds due to $y + 1 < e^y$.

$\square$

