# OpenReview forum: "Should Under-parameterized Student Networks Copy or Average Teacher Weights?"
_NeurIPS.cc/2023/Conference — NeurIPS 2023 poster_

### Official Review · Reviewer_Zuqd · 2023-07-08

**Soundness:** 3 good
**Presentation:** 2 fair
**Contribution:** 2 fair
**Rating:** 4
**Confidence:** 2

**Summary:**

This paper is a theoretical study of whether an under-parameterized student network should copy or average teacher weights, there teacher is a one-hidden-layer network with k hidden units and student is the same but with n<k hidden units. The paper derived the optimal “copy-average” configuration under a set of conditions, and also provided a closed-form solution for the case of n=1.

**Strengths:**

derived some theoretical results in terms of whether smaller two-layer student network should copy or average the weight of a larger two-layer teacher network under certain conditions


**Weaknesses:**

The conditions for the results to hold seems to be very strict, far from practical. I struggle to understand the practical significance of this theory

intuition is not well explained

mathematical derivation is presented in a straightforward way


**Questions:**

Why are the theoretical results presented important when there are so many unrealistic assumptions?

Why would one “expect that the incoming and outgoing weights of a student neuron are either identical to those of a 29 teacher neuron or that they are closely aligned with the weights of a group of teacher neurons”? What’s the rationale of this expectation?

Could you elaborate how “the loss function in Eq. 5 can be expanded as a weighted sum of 109 the following Gaussian integral terms”, and further expressing “the integral in terms of the covariance of the two-dimensional Gaussian”


**Limitations:**

practical significance of the results

---

> ### Author Rebuttal · Authors · 2023-08-09
>
> We thank the reviewer for their comments. Below, we address each point. We hope that the reviewer will find the results more intuitive. From a bird-eye view, our main contribution is the copy-average phenomenon that kicks in for underparameterized networks. On practical relevance, the results suggest a novel and applicable method to distill a large (teacher) neural network into a smaller (student) network for budget-limited scenarios which we discuss below.
>
> **Why is the under-parametized regime interesting in practice?** While it is true that the theory of deep learning community has focused on overparameterized regime in recent years, we do not think that this is the only regime of interest. Two reasons: (1) the global optimum of the under-parameterized networks creates symmetry-induced critical points in overparameterized networks [8], shaping their loss landscape and learning dynamics, (2) how to optimally compress a large (teacher) network into a smaller (student) network is an open question and we give a recipe to approach this problem. While it is true that our theorems hold for a limited setting as usual for a theory paper, one may expect that the copy-average phenomenon holds more generally. For distillation, it is advisable to initialize the (student) network by copying $n-1$ teacher neurons & averaging the remaining teacher neurons, and then training it further. The copy-average insight might be the key to finding the optimal solution and help improve the distillation methods. We will add a discussion on distillation in the conclusion.
>
> **Intuition is not well explained:** We think this comment is unwarranted. Recall the main theorem first: in the under-parameterized regime with $n \ll k$, the loss landscape exhibits various non-global critical points, and the global optimum is achieved when $n-1$ student neurons copy $n-1$ teacher neurons each and the last student neuron implements an average of the remaining teacher neurons. We prove this by using the analytic expression of the optimal loss of one-neuron network denoted by $L^*_\text{erf}(k)$ (Eq. 16). In particular, because $L^*_\text{erf}(k)$ is (discrete) concave, we get Lemma 5.4 which yields the copy-average optimality of the $(n-1)$-copy-$1$-average configuration. This is all to say that this result is non-trivial and comes from algebraic properties. We even take a further step to say that our results are not necessarily intuitive, but this is not negative by any means. New results that are unintuitive at the time they are published in fact might push the boundaries of fundamental sciences. Figure 1 is simplified as well (see the common response to above).
>
> **Assumptions are far from practice:**
> * *Input data is standard Gaussian:* See the common response above.
> * *The teacher network is orthonormal:* See the common response above.
>
> **...the rationale for either copy or average groups of teacher neurons:** The optimal solution of the one-neuron (student) network that approximates a multi-neuron teacher network is to ‘average’ all teacher neurons as we prove in Section 5.1. Once this is known, one might expect that the student neurons either exactly –copy– teacher neurons, or –average– groups of teacher neurons as done by the optimal one-neuron network. Even if the reader did not have this expectation, we suggest forming such an expectation early on to warm up the reader to the results of the paper.
>
> **Weighted sum of 109 via the Gaussian integral term/expressing it in terms of covariance:** Let us break this down. Eq 5. can be expanded as a sum of squares (since it is a square of a sum of neurons) in terms of summands
>
>  * $a_1 a_2 \mathbb{E}[\sigma(w_1 x) \sigma(w_2 x)] $ (student-student term)
>  * $a_1 b_2 \mathbb{E}[\sigma(w_1 x) \sigma(v_2 x)] $ (student-teacher term)
> * $ b_1 b_2 \mathbb{E}[\sigma(v_1 x) \sigma(v_2 x)] $ (teacher-teacher term).
>
> One such term is written in Eq. 109 and $w_1$ and $w_2$ denote arbitrary vectors here. We understand it might be confusing here as $w_1$ and $w_2$ denote also student incoming vectors, hence we will write $V_1$ and $V_2$ instead in the revised version for two arbitrary vectors that represent (a) $(w_1, w_2)$, or (b) $(w_1, v_2)$, or (c) $(v_1, v_2)$.
>
> The Gaussian integral term of Eq 109 is of the form $\int f(x, y) p(x, y) dx dy$ where $p$ is a two-dimensional Gaussian density function. One can always reparameterize this integration after integrating out $x$ and $y$ in terms of the mean and covariance of the two-dimensional Gaussian (equivalently, the parameters of the density function $p$). Therefore, it is possible to express it in terms of the mean (which is zero) and $2 \times 2$ covariance (shown in Eq in line 111). We are using a specific parameterization of the covariance in terms of $r_1$ ($\ell_2$-norm of $w_1$), $r_2$ ($\ell_2$-norm of $w_2$), and $u$ (normalized dot product of $w_1$ and $w_2$). This yields the so-called interaction function given in Eq 7. We’ll update the text here by changing to slack variables and adding explanations on expressing the integral with covariance parameters.

---

> > ### Author Response · Authors · 2023-08-15
> >
> > Dear Reviewer ```Zuqd```,
> >
> > Did you find the time to look at the responses? Since the reviewer-author discussion time is running out, we would like to hear from you.

---

> > ### Comment · Reviewer_Zuqd · 2023-08-17
> > **empirical results?**
> >
> > thanks for the detailed response! Regarding "For distillation, it is advisable to initialize the (student) network by copying
> >  teacher neurons & averaging the remaining teacher neurons, and then training it further", I probably shouldn't ask about experiments for a theory paper, but since the theory suggests practical algorithm to improve distillation, any empirical results?

---

> > > ### Author Response · Authors · 2023-08-18
> > >
> > > Thank you for replying. Empirical results on the practical application of our theory to improve distillation are beyond the scope of this paper.
> > >
> > > If the rebuttal alleviated your concern ```I struggle to understand the practical significance of this theory``` we invite you to reconsider your evaluation. We are not aware of any criteria that Neurips theory papers should *demonstrate* empirical relevance. But we agree that this would be an interesting question for future research.

---

### Official Review · Reviewer_ThaK · 2023-07-13

**Soundness:** 3 good
**Presentation:** 3 good
**Contribution:** 3 good
**Rating:** 6
**Confidence:** 4

**Summary:**

The paper studies a shallow teacher-student setting in which a student network being trained attempts to recover the orthogonal weights of a teacher network, assuming normally distributed data. Specifically, the setting being studied is where the number of student neurons $n$ is far smaller than that of the teacher, $k$. An analysis of the optimization landscape is given for the case of a single student neuron, and assuming an odd activation function, it is shown that among the family of critical that are combinations of averages of the teacher's neurons (CA points), the optimal such point is the one where $n-1$ student neurons copy $n-1$ teacher neurons, and the last student neuron averages the remaining teacher neurons.

While I'm overall impressed by the technical results in the paper and how well it is written, I find the results a bit limited. On the other hand, this paper studies a technical and challenging problem, and with this in mind, I do believe its merits outweigh its drawbacks.


Post-rebuttal:

I have decided to increase my score following the rebuttal and my discussion with the authors. My overall opinion of the paper remains positive and I'm leaning more towards acceptance.

**Strengths:**

- The paper is well structured and written.

- The problem studied is very technical and making interesting observations on it is challenging.

- The technical idea behind the paper (reformulation as a constrained optimization problem) seems interesting and innovative.

**Weaknesses:**

- The results provided in the paper only partially solve the problem it addresses. For example, the optimality of the copy $n-1$ and average $k-n+1$ teacher neurons point is only established within the family of CA points, and it's possible that there are other non CA points which achieve an even better loss. While a more thorough understanding of the single neuron case is provided, it is of course still limited since it's just a single neuron.

- The analysis only applies to the under parameterized regime where potentially the class of student networks is not even capable of expressing a remotely useful solution. It is far more interesting to study the regime where there are local minima that can achieve small (but not necessarily zero) loss, since this regime might have stronger practical implications. Moreover, the orthogonality assumption which implies $k\leq d$ further restricts this setting to $n\ll k\leq d$, which needless to say does not capture a setting similar to how neural networks are trained in practice in terms of the magnitudes of these quantities.

**Questions:**

- Throughout the paper, in the experimental parts, it is mentioned that gradient flow converges to certain CA or close to CA points. How can you simulate gradient flow? Did you happen to use a very small step size gradient descent? If so please clarify this, since this doesn't imply that gradient flow will necessarily exhibit the same behavior.



Minor comment:

Line 266: "span of teacher's" -> "span of the teacher's"

**Limitations:**

Yes.

---

> ### Author Rebuttal · Authors · 2023-08-09
>
> We thank the reviewer for the actionable feedback. Thanks also for acknowledging the fundamental difficulty of the teacher-student problem. Below, we respond to each feedback. Some of the results cover the full under-parameterization regime (n<k) which is much more general than what is understood by the reviewer ($n \ll k$). We appreciate follow-up questions, if there is anything unclear or any new questions arise.
>
> **I find the results a bit limited:** The unit-orthonormal teacher and standard Gaussian input data are common assumptions (see the common response above). Whereas the ‘realizable’ case when $n=k$ is well-studied for example by Saad&Solla 1995 and Safran&Shamir 2018, the under-parameterized regime $n<k$ is not studied so far. Importantly, the optimal CA-point is a $(n-1)$-copy-$1$-average critical point in the full under-parameterized regime (for any $n<k$).
>
> What is special about the $n \ll k$ regime, is the empirical observation that gradient flow converges to the optimal CA critical point (see Figure 4). However, this is not the end of the story as we show in Figure 4: gradient flow converges to another critical point with lower loss in the red regime when n is close to k. Based on this, we conjecture that the global optimum is in fact the CA-optimum when $n \ll k$ and propose a constrained optimization formulation. We believe the different regimes found here are intriguing and one may need to develop new proofs to find the global optimum in the full under-parameterization regime. **The results only partially solve the problem…** The system shows an abrupt change in behavior (akin to a phase transition) at about $n \sim 0.8 k$ (red regime in Figure 4): the gradient flow converges to the perturbed-$n$-copy configuration for all $20$ seeds. Such systems are known to be challenging, for example, for the Ising model, the mathematical development of even the ‘simple’ models has taken 100 years. No one knows at this point how difficult it is to complete the theory of shallow teacher-student learning.
>
> **Why the problems with and without local minimum are *both* interesting:** While we agree with the reviewer that problems with various local minima show richer phenomena, so far a general mathematical treatment of such problems was not possible for neural networks. The problems without local minima are quite interesting as well as we argue next. *Problems without local minimum:* Let us consider linear networks as a case study. Without explicit regularization, it is well known that there is no local minimum [S1]. However, the training dynamics is non-trivial and this is still an active field of research [S2,S3,S4,S5,S6]. *Problems with local minima:* The empirics show that there is a regime of underparameterization (see Figure 4-B2, gray regime) where the gradient flow converges to either $(n-1)$-copy-$1$-average minimum or (ii) perturbed-$n$-copy minimum.
>
> **...further restricts this setting to $ n \ll k \leq d $:** To put it simply, we hypothesize that this is the simplest but generic problem in shallow learning and it is an essential step towards more realistic modes. Let us break down this hypothesis.
>
> *(1) How to drop the orthogonality assumption?* See Remark C.1, line 475 in the Appendix. In short, the constrained optimization trick applies to non-orthogonal teacher networks as well, up to a change in the constraints by writing them on the basis of the teacher’s incoming vectors. However, one needs linear independence which still imposes k≤d. The analysis of linearly dependent teacher incoming vectors may invoke yet additional constraints in the formulation.
>
> *(2) Are the results restricted to n≪k?* Once again, no. The optimal copy-average point is $(n-1)$-copy-$1$-average for all $n<k$ due to Lemma 5.4 (and the paragraph after). However, as written above and in Figure 4, there are other minima found by gradient flow when $n$ is slightly bigger than $0.5 k$. Hence, the conjecture that the global optimum is the CA-optimum is stated for the limited case $n \ll k$ and proven for $n=1$. With more effort and clever methods, we believe the constrained optimization problem can be solved for $n=2$ where it is exact (it is a relaxation starting from $n=3$, see Appendix C.3).
>
> **How did we simulate gradient flow?** We used the MLPGradientFlow package implemented by [6] (https://github.com/jbrea/MLPGradientFlow.jl). We used the default option in the package that calls the KenCarp58 ordinary differential equation (ODE) solver from Julia, which is different from implementing gradient descent with a small step size. The package features analytic formulas of the integrals for erf and ReLU for accurate computations of gradients and Hessians. Training is done for 1e5 ode iterations. For erf, all seeds converged to configurations with gradient norm below 1e-10 (see Appendix B lines 416-428, and [6] for more details).
>
> [S1] Baldi, Pierre, and Kurt Hornik. "Neural networks and principal component analysis: Learning from examples without local minima." Neural networks 2.1 (1989): 53-58.
> [S2] Li, Zhiyuan, Yuping Luo, and Kaifeng Lyu. "Towards resolving the implicit bias of gradient descent for matrix factorization: Greedy low-rank learning." arXiv preprint arXiv:2012.09839 (2020).
> [S3] Jacot, Arthur, et al. "Saddle-to-saddle dynamics in deep linear networks: Small initialization training, symmetry, and sparsity." arXiv preprint arXiv:2106.15933 (2021).
> [S4] Pesme, Scott, Loucas Pillaud-Vivien, and Nicolas Flammarion. "Implicit bias of sgd for diagonal linear networks: a provable benefit of stochasticity." Advances in Neural Information Processing Systems 34 (2021): 29218-29230.
> [S5] Even, Mathieu, et al. "(S) GD over Diagonal Linear Networks: Implicit Regularisation, Large Stepsizes and Edge of Stability." arXiv preprint arXiv:2302.08982 (2023).
> [S6] Pesme, Scott, and Nicolas Flammarion. "Saddle-to-Saddle Dynamics in Diagonal Linear Networks." arXiv preprint arXiv:2304.00488 (2023).

---

> > ### Author Response · Authors · 2023-08-11
> > **We will add a discussion on the limitations**
> >
> > We will add a discussion on the limitations as well based on ```ThaK``` s and other reviewers comments:
> >
> > * The constrained optimization problem is exact only for $n=1$ and $n=2$. We will discuss on the gap between the constrained optimization and the original problems and how one might close the gap (see the response to ```6p1W```). In this sense, the exact constrained optimization formulation is limited to $n \ll k$.
> >
> > * On the assumption $k \leq d $: This prohibits us from studying more realistic teacher models that comply with the universal approximation theorem, and we will state it clearly.
> >
> > We wanted to add this comment since it might not have been clear from the official rebuttal. We think this should **completely** resolve ```ThaK``` s point on unaddressed limitations.

---

> > > ### Author Response · Authors · 2023-08-15
> > >
> > > Dear Reviewer ```ThaK```,
> > >
> > > Did you find the time to look at the responses? Since the reviewer-author discussion time is running out, we would like to hear from you.

---

> > > > ### Comment · Reviewer_ThaK · 2023-08-18
> > > > **Post-rebuttal response**
> > > >
> > > > Dear authors,
> > > >
> > > > Thank you for your detailed response. I appreciate your intensions to incorporate some of my suggestions.
> > > >
> > > > Regarding the simulation of gradient flow, I believe a short explanation of how you simulated it (i.e. that you computed an approximation of the path along which the gradient is never larger than $10^{-10}$) would help clarify your claims.
> > > >
> > > > Lastly, regarding the significance of the contributions in the paper, I still find the setting of the problem studied somewhat limiting, and I stand behind my statement that the setting where $n\ll k\leq d$ is not as interesting as the case $n\geq k$. However, I agree with you that the problem being studied is interesting and that your results are of interest.

---

> > > > > ### Author Response · Authors · 2023-08-19
> > > > >
> > > > > Dear Reviewer ```ThaK```,
> > > > >
> > > > > Thank you for the response! We shall incorporate how we implemented the gradient flow following your suggestion.
> > > > >
> > > > > Let us emphasize once again our results are not limited to $n \ll k$ and pertain to the whole under-parameterized regime ($n < k$). See the response above to the criticism *"I find the results a bit limited."* Indeed, we do not study overparameterization ($n \geq k$) in this paper.
> > > > >
> > > > > If the rebuttal responds to your criticism of the limitations and you think the paper will interest the NeurIps community, we invite you to consider improving your score. Thank you!

---

### Official Review · Reviewer_6p1W · 2023-07-19

**Soundness:** 3 good
**Presentation:** 2 fair
**Contribution:** 3 good
**Rating:** 6
**Confidence:** 3

**Summary:**

his paper proposes a constrained optimization approach to learning
student networks $g(x) = \sum_{j=1}^n a_j \sigma (v_j \cdot x)$ that
have fewer neurons than a two-layer teacher network $f^* (x) =
\sum_{i=k} b_i \sigma( w_i \cdot x )$. (with $ n \le k $ as the student network has
fewer neurons than the teacher )

Starting with the squared-loss between the student and teacher, the authors
propose a relaxed constrained optimization problem in terms of the student parameter norms
and correlations to themselves and the teacher parameters.
The authors show that in the case
of general $b$ and odd-activations, for any $n$-partition of the $k$
teacher neurons, if $(a_j^* , v_j^* )$ are a critical point for a
one-neuron student network, then $v_j^*$ lies in the span of the
teacher weights $w_j$ corresponding to the partition. They further
show that for a multi-neuron student network, concatenating the
critical points from one-neuron student networks will produce a
critical point for the overall multi-neuron network problem.

Then the authors jump to unit-norm teacher networks, where $b_i = 1$,
where the critical points are unique, and closed-form solutions exist
for the erf and ReLU activations.

**Strengths:**

- The stated problem is interesting and I find the results compelling.

- There are several non-trivial aspects to the paper. The authors
  first convert the student-teacher minimization problem to a relaxed
  version where they try to optimize variables corresponding to: (a)
  student parameter norms, (b) correlations between student vectors,
  and (c) correlations between student and teacher vectors. As the
  input distribution is Gaussian, the pairwise correlations can be
  analyzed for a wide class of activation functions.  While some of
  the techniques are standard (like odd activation functions to
  decouple neurons in a multi-neuron network), they are compiled well.

- The analysis is fairly slick and the assumptions are reasonable.

**Weaknesses:**

- The constrained optimization is a relaxation of the original
  problem of learning a student network, and the critical points of
  the relaxed problem are analyzed. There is no discussion about the
  gap between the original and relaxed problem. There is some
  discussion about small-scale and medium scale problems in the
  conclusion -- are these related to the gap in the relaxation?

- The teacher network has orthogonal weights, and hence is in a regime
  where the number of neurons is smaller than the input dimension. I'm
  not familiar with the literature on learning student networks -- is
  this standard?

- The authors say that "for larger n that is close to k, we observe a
  transient phase where the gradient flow finds a perturbed copy
  solution that has a lower loss than the optimal CA point." How is
  this possible if your program is a relaxation of the gradient flow
  objective, and why does it not contradict your claim that "For
  shallow neural networks with odd activation function, we prove that
  “copy-average” configurations are critical points, and the optimum
  is reached when $n-1$ student neurons with erf activation each copy
  one teacher neuron and the n-th student neuron averages the
  remaining $k -n +1$ teacher neurons"?  I've complained about this elsewhere, but
  having a Theorem statement for section 5.2 would help alleviate this concern.
  Also, please maintain consistency between erf and odd-activations, as the sentence I
  quoted switches between them.

- The writing can be slightly confusing from time to time because of a
  lot of context switching between one-neuron student networks and
  multi-neuron odd activation function networks, and then to unit-norm
  teacher networks, where the outgoing weights are $b_i = 1$.

- Figure 1 is very hard to parse. How should a reader understand that
  the results in Figure A2 implies convergence to a copy-average of
  the true solution? And why are the blocks for the
  pairwise-correlation $u_1$ and $u_2$ weird rectangular blocks?
  Wouldn't the pairwise correlations just be $(1, \rho, \rho, 1)$?
  Overall, A2, A3, A4 are supposed to explain different settings for
  $n, k$, but in their current form, they are all equally
  incomprehensible. The caption is also half a page. There's too much
  going on, it needs to be simplified.

- Why is there no theorem statement for section 5.2? It seems like a
  strange choice to not include it given that it's your main result.
  The lemma currently stated is for a two-neuron network.

- Minor:

   - The first line of the abstract and introduction says that any
     continuous function can be approximated arbitrarily well given
     enough neurons in the teacher network, which is true. However,
     all the results in this paper can only hold when the number of
     neurons ($k$) is atmost the dimension of the input ($d$), which
     strongly contradicts the first statement.

**Questions:**

See points 1, 2, 3 in the weaknesses section above. Overall, I like this paper, but I feel that the presentation makes it hard to parse the results and some statements seem contradictory.

**Limitations:**

Could be discussed in more detail. It's merged with the conclusions section as a monolithic paragraph. The gap between the relaxation and original problem are not discussed.

---

> ### Author Rebuttal · Authors · 2023-08-09
>
> We thank the reviewer for the detailed feedback. Thanks for liking the results too! Below, we aim to address each one of the concerns. We are confident that the writing can be improved with minor updates on the text, following the suggestions of the reviewers. This would be a minor change overall since the mathematics is stable (i.e. nothing is contradictory). For the discussion and the conclusion parts, we will structure them better using the extra page.
>
> **The gap between the constrained optimization and the original problem:** First, the constrained optim. problem is in fact exactly equivalent to the original for $n=1$ and $n=2$. Because the mapping from the weight space to the order-parameter space is invertible in this case. We exploit this to prove that the global optimum is an ‘average’ neuron for $n=1$. However, for $n \geq 3$, due to the geometry of the problem, the mapping is no longer invertible. In particular, we gave an example configuration in the order parameter space that is not in the image of this map in Appendix C.3 (on page 18). Overall, at the moment, we are exploiting the constraints for pairwise angles by Cauchy-Schwartz, however, it may be possible to tighten these constraints (hence closing the gap between the two optimization problems) by exploiting the geometrical constraints of angle constellations between triplets or quadruples of vectors. We will add this discussion on the gap between the two problems in the paper.
>
> **Regarding the perturbed-n-copy solution that has a lower loss than the optimal CA point:** Let us break down the sentence that the reviewer has quoted “For a shallow network with odd activation…”. Here we are referring to the ‘copy-average optimum’. More precisely, the statement is: among the copy-average configurations, for a unit-orthonormal teacher, for the erf activation function, the optimum (copy-average) configuration is an $(n-1)$-copy-$1$-average point. This is our main result. The question is then: is the CA optimum also the global optimum? Empirically, gradient flow converges to the CA-optimum for $n \ll k$ and we conjecture that the answer is positive in this regime. However, for $n \gtrsim 0.8k$, gradient flow converges to another critical point with lower loss, which we call perturbed-n-copy. This is an intriguing phenomenon: the system seems to exhibit abrupt changes in behavior (green-grey-blue-red regimes) as we increase n. Taken together, proving the exact parameters of the global optimum of under-parameterized shallow networks might be a very challenging optimization problem and it is beyond the scope of this paper.
>
> **Orthogonal teacher/Relevant literature:** See the common response above.
>
> **The writing can be slightly confusing:** In Section 3, we present a constrained optimization formulation of the classic teacher-student problem. In Section 4, we focus on odd activation functions and orthogonal teachers and prove that concatenating critical points of the smaller networks yields new critical points. In Section 5, we focus on the unit-orthonormal teacher and we have two subsections. In Section 5.1, we have some properties of the exact parameters of the global optimum for $n=1$ for general activation functions, and a corollary (5.2) where we give the exact parameters for erf. In Section 5.2, we focus on the erf activation function and prove that the optimum CA-point is an $(n-1)$-copy-$1$-average point. To help the reader, we will add an overview with a bullet point list of contributions that highlights the assumptions of the main results.
>
> **Why is there no Theorem in Section 5.2?** We wanted to present the proof in a constructive way and that’s why we put Lemma 5.4 in the main. This constitutes the big chunk of the proof (of the statement that the optimum CA-point is an $(n-1)$-copy-$1$-average) and the remaining argument is written in lines 240-246. We understand how this might be confusing. Therefore, we will instead include the main result as a theorem here as suggested by the reviewer, and send the Lemma to the Appendix for the interested reader.
>
> **Regarding Figure 1:** See the common response above. In addition, in A2-top (for erf), we have $u_1=(1, 0, 0)$ and $u_2=(0, 1/\sqrt{2}, 1/\sqrt{2})$ which shows that (1) the two student vectors are orthogonal to each other and (2) the first neuron is a copy-neuron and the second neuron is an average neuron. For ReLU (A2-bottom), there is no perfect decoupling (see the non-zero off-diagonals), however, the correlation matrix is similar to erf. We removed B to simplify it and made the caption shorter. Please see the updated figure in the pdf above. Is it understandable now? If not, we’d greatly appreciate follow-up feedback. We’d like to make it easily digestible for the general ML community.
>
> **Minor Comment:** Thank you for raising this point. We will update the abstract and the first sentence of the introduction accordingly and remove the reference to the universal approximation theorem.

---

> > ### Author Response · Authors · 2023-08-15
> >
> > Dear Reviewer ```6p1W```,
> >
> > Did you find the time to look at the responses? Since the reviewer-author discussion time is running out, we would like to hear from you.

---

### Official Review · Reviewer_Cg9m · 2023-07-24

**Soundness:** 3 good
**Presentation:** 2 fair
**Contribution:** 3 good
**Rating:** 7
**Confidence:** 3

**Summary:**

The goal of the paper is to understand the following approximation problem: given a student network and teacher network, how should one construct the student network such that the approximation error is minimized. The paper considers the underparameterized regime where the student network has fewer parameters than the teacher network. The first half of the paper considers a constrained optimization problem which is equivalent to the original approximation problem if the student network consists of one neuron (otherwise, the constrained optimization problem is a relaxation of the original approximation problem). The second half provides a characterization of a subset of critical points for the approximation problem. Given additional assumptions on the activation function used in the networks, they also provide an analytical formula for non-trivial critical points of the approximation problem and some empirical verification of the paper's claims.

**Strengths:**

The paper is mostly well written. In addition, to the best of my knowledge, the proof (in restricted settings) and the empirical observations (in more general settings) of the copy average of phenomenon seems to be novel. Furthermore, the constrained optimization problem that serves as a relaxation of the original approximation problem is interesting. In addition, the paper provides a nice family of critical points for the approximation problem that can be constructed by concatenating critical points of smaller approximation problems.

**Weaknesses:**

Many of the results of the paper requires that the student network consist of one neuron. While section 5.2 gives a characterization of critical points for the multi-neuron case, it requires that the activation function is erf, an uncommon activation function. In addition, the input data is assumed to be gaussian.

**Questions:**

In the context of section 5.2 multi-neuron network, is there a nice way to characterize which of the n - 1 teacher neurons will be copied other than trying every (n - 1) copy configurations and computing their approximation error?

For figure 1.A2, it is not clear which pair of student and teacher neurons are being used to compute the correlation (perhaps label the y-axis with the teacher neurons v1, ..., vk). Furthermore, I don't really see what insight the plots of r_i and a_i give (perhaps add corresponding teacher r'_i and a'_i ? )

Does lemma 5.4 hold for other activation functions? This seems to be the major workhorse for characterizing critical points for student networks for multiple neurons and thus it would be nice to address if a similar lemma holds for common activation functions.

---

> ### Author Rebuttal · Authors · 2023-08-09
>
> We thank the reviewer for the feedback and appreciating the novelty of the copy-average phenomenon. Below, we answer every question raised by the reviewer. We appreciate follow-up questions, if anything is unclear in the responses or if the reviewer has new questions.
>
> **Which of the $n-1$ teacher neurons will be copied?** In the teacher model used in the paper and in the literature, all outgoing weights are one and the incoming vectors are orthogonal (i.e. unit-orthonormal teacher). Because of this symmetric arrangement, all copy-average optima, where one of the $n-1$ teacher neurons is copied, have the same loss. It is therefore impossible to predict which teacher neurons will be copied.
> This symmetry is broken, if the outgoing weights are arbitrary or the incoming vectors are not orthonormal. In this case, copying one teacher neuron may have a lower loss than copying other teacher neurons. Although this is beyond the scope of this paper, one can probably use the tools developed here to tackle this difficult question. First, one needs to compute the approximation error of the one-neuron network. This requires solving Eq. 15, where the first summand would have an additional $b_j$ factor. $u_j$’s will no longer be equal but they will solve a similar analytic equation. Once $L^*_\text{erf}(b_1, …, b_k)$ is obtained, we need a more general version of Lemma 5.4. This is an interesting question!
>
> **(Regarding Figure 1) Furthermore, I don't really see what insight…:** See the common response above for Figure 1. The teacher is unit-orthonormal, i.e. $a_i’=1, r_i’=1$, we added it in the caption. For erf (top row), $a_i$ and $r_i$ exactly match the theory ( bars-empirics and dashed lines-theory overlap). For ReLU (bottom row), we think it is interesting to see (1) $r_i$ and $a_i$ are very close to each other (aka balancedness condition, though we did not force balancedness at initialization by small initialization or brute force) (2) all student neurons have similar `magnitudes’, that is all $r_i$ and $a_i$ are also very close to each other. We think these observations will be helpful in the future for completing the theory for ReLU.
>
> **Does Lemma 5.4 hold for other activation functions?** Lemma 5.4 follows from the concavity of the optimal one-neuron loss, that is $L^*_\text{erf}(k)$. It is possible to compute it for other activation functions. For odd monomial activation functions $\sigma(x)=x^b$ (b odd), the interaction function $g$ admits an analytic formula. Following the same recipe, one can simply check whether $L^*_b(k)$ is concave. However, this class of activation functions is uncommon in practice. If the reviewer thinks this setting is interesting/helpful/useful, we are happy to include it in our paper. We already have the optimal one-neuron loss for ReLU, Eq. 116, page 28 in Appendix, however, ReLU is not an odd activation function (i.e. Theorem 4.2 does not hold).
> More in detail, for ReLU, the copy-average phenomenon is tricky because (1) the concatenation of critical points is not a new critical point (see Eq 44 in the Appendix), and (2) the off-diagonal correlations are non-zero (see Figure 1). A challenge here is computing the off-diagonal perturbations to the copy-average configurations.
>
> **Erf is uncommon…:** We see how the assumption on the activation function might feel rather limiting. However, we believe, the distillation of this phenomenon into the analytically tractable setting of erf, is a success. We show empirically that the copy-average phenomenon kicks in at least for ReLU (see Appendix B.3, Figure 9) and expect that it is a universal phenomenon. This universality hypothesis is very interesting! It seems that the relevant scaling is when both n and k go to infinity with a linear ratio, however, the theoretical analysis of this setting likely needs a combination of tools from random matrix theory and dynamical systems. We aim to study this challenging but intriguing problem in the future.
>
> Moreover, observe that for ReLU (Figure 9 in Appendix B.3), when both n and k are large, the off-diagonal correlations are small as indicated by lighter colors. This observation supports universality in the limit when both n and k are large which in turn implies that the analysis of erf is in fact generic. Finally, we invite the reviewer to see further literature review in Appendix A, Table 1. All these influential works considered Gaussian data and fixed activation functions to be either erf or ReLU because both cases require distinct analyses. We are able to develop a unifying framework via interaction functions (see Assumptions 3.1) for the analysis of the one-neuron network (see the discussion in Appendix A).
>
> **Input data is standard Gaussian:** see the common response above.

---

> > ### Author Response · Authors · 2023-08-15
> >
> > Dear Reviewer ```Cg9m```,
> >
> > Did you find the time to look at the responses? Since the reviewer-author discussion time is running out, we would like to hear from you.

---

> > > ### Comment · Reviewer_Cg9m · 2023-08-19
> > > **reply**
> > >
> > > Thank you for your response. Upon reflection, I have raised the score from 6 to 7.

---

### Author Rebuttal · Authors · 2023-08-09

**Regarding Figure 1 (response to ```Cg9m```  and ``` 6p1W``` ):** $u_{ij}$ is the vector of correlations between the student incoming vector $w_i$ and the teacher incoming vectors $v_j$, i.e. $u_i = (u_{i1}, …, u_{ik})$. This notation is introduced in line 138 and added in the new caption (see the pdf attached). The last row of the correlation matrix (rectangle) in A2 was gray (in fact, effectively zero). This shows that $w_i$ is in the span of the teacher's incoming vectors $v_1, …, v_k$. We realized too much was going on in this figure. Hence we removed panel B and the last gray row and also simplified the caption (see the pdf attached for the updated Figure 1).

**On the practical relevance of the Gaussian input data (response to ```Cg9m```  and ``` Zuqd```):** For general data distributions, there is no theory framework to do a fine-grained analysis of neural network learning apart from special cases such as the infinite-width limit where the neural tangent kernel or mean-field analysis applies. This is a commonly used assumption (see [1,13,17,20,22,23,25] cited in the paper) and it is natural in high dimensions due to universality (that is, a generic data distribution can be replaced by a Gaussian distribution without affecting measurable quantities of interest) [S1, S2, S3, S4].

**Orthogonal teacher/Relevant literature (response to ```6p1W```, ```ThaK```, and ```Zuqd``` ):** This assumption is standard in the theoretical analysis of the teacher-student setup and we refer the reviewers to Table 1 (Appendix A, page 12). Two options: either the input dimension approaches infinity (in statistical mechanics or in high-dimensional probability approaches) [2,13,20] or the teacher network is assumed to be unit-orthonormal [1,22,23]. Then the incoming vectors of the teacher are either orthogonal or ‘approximately’ orthogonal due to the fact that random vectors are effectively orthogonal in high dimensions (i.e. $d \to \infty$).

Following the literature, we started by analyzing the unit-orthonormal teacher network that yields a very symmetric problem. Different from the literature, our approach is generalizable to general teachers (we can choose arbitrary $b_i$ as done in Proposition 4.1) and it is possible to write the constrained optimization formulation for non-orthogonal incoming vectors (see Remark C.1, line 475 in Appendix). Overall, (1) unit-orthonormal teacher is a commonly studied data-generating model and shows intriguing and non-trivial properties (see Figure 4) (2) the constrained optimization formulation is generalizable to more general teachers, however, the measurable properties of the system will likely be more complicated.

[S1] El Karoui, Noureddine. "The spectrum of kernel random matrices." (2010): 1-50.

[S2] Hu, Hong, and Yue M. Lu. "Universality laws for high-dimensional learning with random features." IEEE Transactions on Information Theory 69.3 (2022): 1932-1964.

[S3] Montanari, Andrea, and Basil N. Saeed. "Universality of empirical risk minimization." Conference on Learning Theory. PMLR, 2022.

[S4] Dandi, Yatin, et al. "Universality laws for Gaussian mixtures in generalized linear models." arXiv preprint arXiv:2302.08933 (2023).

---

### Decision · Program_Chairs · 2023-09-21

**Decision:**

Accept (poster)

**Comment:**

This paper studies theoretical and empirically studies network training in the setting where labels come from a teacher of higher width; theoretically it is shown that for an orthogonal teacher and gaussian data, weights do a copy-average procedure, and this is verified empirically in more general settings. Reviewers are overall positive, with some concerns about the strength of the assumptions; we urge the authors to use the new camera ready space to polish their writing and help convey their story.